# SIGNSGD VIA ZEROTH-ORDER ORACLE

**Sijia Liu**[†]    **Pin-Yu Chen**[†]    **Xiangyi Chen**[‡]    **Mingyi Hong**[‡]
[†]MIT-IBM Watson AI Lab, IBM Research
[‡]University of Minnesota, Twin Cities

## ABSTRACT

In this paper, we design and analyze a new zeroth-order (ZO) stochastic optimization algorithm, ZO-signSGD, which enjoys dual advantages of gradient-free operations and signSGD. The latter requires only the sign information of gradient estimates but is able to achieve a comparable or even better convergence speed than SGD-type algorithms. Our study shows that ZO-signSGD requires $\sqrt{d}$ times more iterations than signSGD, leading to a convergence rate of $O(\sqrt{d}/\sqrt{T})$ under some mild conditions, where $d$ is the number of optimization variables, and $T$ is the number of iterations. In addition, we analyze the effects of different types of gradient estimators on the convergence of ZO-signSGD, and propose several variants of ZO-signSGD with $O(\sqrt{d}/\sqrt{T})$ convergence rate. On the application side we explore the connection between ZO-signSGD and black-box adversarial attacks in robust deep learning. Our empirical evaluations on image classification datasets MNIST and CIFAR-10 demonstrate the superior performance of ZO-signSGD on the generation of adversarial examples from black-box neural networks.

## 1  INTRODUCTION

Zeroth-order (gradient-free) optimization has attracted an increasing amount of attention for solving machine learning (ML) problems in scenarios where explicit expressions for the gradients are difficult or infeasible to obtain. One recent application of great interest is to generate prediction-evasive adversarial examples, e.g., crafted images with imperceptible perturbations to deceive a well-trained image classifier into misclassification. However, the *black-box* optimization nature limits the practical design of adversarial examples, where internal configurations and operating mechanism of public ML systems (e.g., Google Cloud Vision API) are not revealed to practitioners and the only mode of interaction with the system is via submitting inputs and receiving the corresponding predicted outputs (Papernot et al., 2017; Liu et al., 2017; Chen et al., 2017; Tu et al., 2018; Ilyas et al., 2018b; Cheng et al., 2018; Bhagoji et al., 2018). It was observed in both white-box and black-box settings[1] that simply leveraging the sign information of gradient estimates of an attacking loss can achieve superior empirical performance in generating adversarial examples (Goodfellow et al., 2015; Madry et al., 2018; Ilyas et al., 2018a). Spurred by that, this paper proposes a zeroth-order (ZO) sign-based descent algorithm (we call it 'ZO-signSGD') for solving black-box optimization problems, e.g. design of black-box adversarial examples. The convergence behavior and algorithmic stability of the proposed ZO-signSGD algorithm are carefully studied in both theory and practice.

In the first-order setting, a sign-based stochastic gradient descent method, known as signSGD, was analyzed by (Bernstein et al., 2018; Balles & Hennig, 2017). It was shown in (Bernstein et al., 2018) that signSGD *not only* reduces the per iteration cost of communicating gradients, *but also* could yield a faster empirical convergence speed than SGD (Kinga & Adam, 2015). That is because although the sign operation compresses the gradient using a single bit, it could mitigate the negative effect of extremely components of gradient noise. Theoretically, signSGD achieves $O(1/\sqrt{T})$ convergence rate under the condition of a sufficiently large mini-batch size, where $T$ denotes the total number of iterations. The work in (Balles & Hennig, 2017) established a connection between signSGD and Adam with restrictive convex analysis. Prior to (Bernstein et al., 2018; Balles & Hennig, 2017), although signSGD was not formally defined, the fast gradient sign method (Goodfellow et al., 2015) to generate white-box adversarial examples actually obeys the algorithmic protocol of signSGD. The effectiveness of signSGD has been witnessed by robust adversarial training of deep neural networks (DNNs) (Madry et al., 2018). Given the advantages of signSGD, one may wonder if it can

---

[1] 'white-box' (vs 'black-box') implies whether the knowledge on the target model is known *a priori*.

be generalized for ZO optimization and what the corresponding convergence rate is. In this paper, we answer these questions affirmatively.

**Contributions**   We summarize our key contributions as follows.

- We propose a new ZO algorithm, 'ZO-signSGD', and rigorously prove its convergence rate of $O(\sqrt{d}/\sqrt{T})$ under mild conditions.

- Our established convergence analysis applies to both mini-batch sampling schemes with and without replacement. In particular, the ZO sign-based gradient descent algorithm can be treated as a special case in our proposed ZO-signSGD algorithm.

- We carefully study the effects of different types of gradient estimators on the convergence of ZO-signSGD, and propose three variants of ZO-signSGD for both centralized and distributed ZO optimization.

- We conduct extensive synthetic experiments to thoroughly benchmark the performance of ZO-signSGD and to investigate its parameter sensitivity. We also demonstrate the superior performance of ZO-signSGD for generating adversarial examples from black-box DNNs.

**Related work**   Other types of ZO algorithms have been developed for convex and nonconvex optimization, where the full gradient is approximated via a random or deterministic gradient estimate (Jamieson et al., 2012; Nesterov & Spokoiny, 2015; Ghadimi & Lan, 2013; Duchi et al., 2015; Gao et al., 2014; Shamir, 2017; Hajinezhad et al., 2017; Ghadimi et al., 2016; Lian et al., 2016; Liu et al., 2018b;c; Wang et al., 2018). Examples include ZO-SGD (Ghadimi & Lan, 2013), ZO stochastic coordinate descent (ZO-SCD) (Lian et al., 2016), and ZO stochastic variance reduced gradient descent (ZO-SVRG) (Liu et al., 2018c;a; Gu et al., 2016). Both ZO-SGD and ZO-SCD can achieve $O(\sqrt{d}/\sqrt{T})$ convergence rate. And ZO-SVRG can further improve the iteration complexity to $O(d/T)$ but suffers from an increase of function query complexity due to the additional variance reduced step, known as 'gradient blending' (Liu et al., 2018c), compared to ZO-SGD. The existing work showed that ZO algorithms align with the iteration complexity of their first-order counterparts up to a slowdown effect in terms of a small-degree polynomial of the problem size $d$.

## 2   SIGNSGD & ITS CONNECTION TO ADVERSARIAL MACHINE LEARNING

In this section, we provide a background on signSGD, together with the problem setup of our interest. In particular, we show that the commonly-used methods for generating adversarial attacks fall into the framework of signSGD.

**Preliminaries on signSGD**   Consider a *nonconvex* finite-sum problem of the form

$$\underset{\mathbf{x}}{\text{minimize}} \quad f(\mathbf{x}) := (1/n) \sum_{i=1}^{n} f_i(\mathbf{x}), \tag{1}$$

where $\mathbf{x} \in \mathbb{R}^d$ are optimization variables, and $\{f_i\}$ are $n$ individual nonconvex cost functions. The finite-sum form (1) encompasses many ML problems, ranging from generalized linear models to neural networks. If the gradients of $\{f_i\}$ are available, then problem (1) can be solved by many first-order methods such as SGD, SCD, and signSGD. The method of our interest is signSGD, which differs from SGD and SCD, takes the *sign* of gradient (or its estimate) as the descent direction. It was recently shown in (Bernstein et al., 2018) that signSGD is quite robust to gradient noise and yields fast empirical convergence.

Algorithm 1 provides a generic sign-based gradient descent framework that encapsulates different variants of signSGD. In Algorithm 1, $\mathrm{GradEstimate}(\cdot)$ signifies a general gradient estimation procedure, which adopts either a *stochastic* gradient estimate in the first-order setting (Bernstein et al., 2018) or a *function difference* based random gradient estimate in the ZO setting (Nesterov & Spokoiny, 2015; Duchi et al., 2015). We call the ZO variant of signSGD 'ZO-signSGD', which will be elaborated on in Sec. 3.

**Adversarial attacks meet signSGD**   It is now widely known that ML models (e.g., deep neural networks) are vulnerable to *adversarial attacks*, which craft inputs (e.g., images) with imperceptible perturbations to cause incorrect classification (Szegedy et al., 2013; Goodfellow et al., 2015; Kurakin et al., 2017; Lin et al., 2019). The resulting inputs crafted by adversaries are known as *adversarial examples*. Investigating adversarial examples not only helps to understand the limitation of learning models, but also provides opportunities to improve the models' robustness (Papernot et al., 2016;

---

**Algorithm 1** Generic sign-based gradient descent

---

1: Input: learning rate $\{\delta_k\}$, initial value $\mathbf{x}_0$, and number of iterations $T$
2: **for** $k = 0, 1, \ldots, T - 1$ **do**
3:      $\hat{\mathbf{g}}_k \longleftarrow \mathrm{GradEstimate}(\mathbf{x}_k)$   # applies to both first and zeroth order gradient estimates
4:      sign-gradient update

$$\mathbf{x}_{k+1} = \mathbf{x}_k - \delta_k \mathrm{sign}(\hat{\mathbf{g}}_k), \quad \text{where } \mathrm{sign}(\mathbf{x}) \text{ takes element-wise signs of } \mathbf{x} \tag{2}$$

5: **end for**

---

Athalye et al., 2018; Madry et al., 2018). In what follows, we show that the generation of adversarial examples in (Goodfellow et al., 2015; Kurakin et al., 2017) can be interpreted through signSGD.

Let $\mathbf{x}_0$ denote the *natural* (legitimate) input of an ML model associated with the true label $t_0$, and $\mathbf{x}' = \mathbf{x}_0 + \boldsymbol{\delta}$ be the *adversarial* example to be designed, where $\boldsymbol{\delta}$ are adversarial perturbations. If $f(\mathbf{x}, t_0)$ is the training loss of a learning model, then the goal of (white-box) adversarial attack is to find minimal perturbation $\boldsymbol{\delta}$ that is sufficient to mislead the learning model, namely, to maximize the loss $f(\mathbf{x}_0 + \boldsymbol{\delta}, t_0)$. Taking the first-order approximation of $f(\mathbf{x}', t_0)$ around $\mathbf{x}_0$, we obtain $f(\mathbf{x}', t_0) \approx f(\mathbf{x}_0, t_0) + \langle \nabla_{\mathbf{x}} f(\mathbf{x}_0, t_0), \boldsymbol{\delta} \rangle$. By constraining the strength of perturbation in the $\ell_\infty$ ball of small radius $\epsilon$ (i.e., $\|\boldsymbol{\delta}\|_\infty \leq \epsilon$), the linear approximation of $f(\mathbf{x}', t_0)$ is then maximized at $\boldsymbol{\delta} = \epsilon \, \mathrm{sign}(\nabla_{\mathbf{x}} f(\mathbf{x}_0, t_0))$ (Shaham et al., 2018). Therefore, generation of adversarial examples proposed in (Goodfellow et al., 2015) obeys the sign-gradient update rule in (2),

$$\mathbf{x}' = \mathbf{x}_0 - \epsilon \, \mathrm{sign}(-\nabla_{\mathbf{x}} f(\mathbf{x}_0, t_0)).$$

Such a connection between adversarial example generation and signSGD also holds in other attacks, e.g., the iterative target attack method (Kurakin et al., 2017). Similarly, a so-called *black-box* attack (Ilyas et al., 2018a; Bhagoji et al., 2018) is associated with our proposed *ZO-signSGD* algorithm.

## 3   ZO-SIGNSGD FOR BLACK-BOX OPTIMIZATION

One limitation of signSGD (Bernstein et al., 2018) is the need of first-order information, i.e., stochastic gradients. However, there exists a large practical demand for solving ML problems where explicit expressions of the gradients are difficult or infeasible to obtain, e.g., the generation of adversarial examples from black-box neural networks as discussed in Sec. 1 and 2.

**Gradient estimation via ZO oracle**   In the ZO setting where the first-order information is unavailable, the gradient estimator at Step 3 of Algorithm 1 has only access to function values of $\{f_i(\mathbf{x})\}$ given a query point $\mathbf{x}$. Based on that, we construct a ZO gradient estimate through a forward difference of two function values (Nesterov & Spokoiny, 2015; Gao et al., 2014; Duchi et al., 2015). In Algorithm 1, $\mathrm{GradEstimate}(\mathbf{x})$ is then specified as

$$\mathrm{GradEstimate}(\mathbf{x}) = \frac{1}{bq} \sum_{i \in \mathcal{I}_k} \sum_{j=1}^{q} \hat{\nabla} f_i(\mathbf{x}; \mathbf{u}_{i,j}), \hat{\nabla} f_i(\mathbf{x}; \mathbf{u}_{i,j}) := \frac{d[f_i(\mathbf{x} + \mu \mathbf{u}_{i,j}) - f_i(\mathbf{x})]}{\mu} \mathbf{u}_{i,j}, \tag{3}$$

where $\mathbf{x} = \mathbf{x}_k$ in Algorithm 1, $\mathcal{I}_k$ is a mini-batch of size $|\mathcal{I}_k| = b$, $\{\mathbf{u}_{i,j}\}_{j=1}^{q}$ are i.i.d. random directions drawn from a uniform distribution over a unit sphere, and $\hat{\nabla} f_i(\mathbf{x}; \mathbf{u}_{i,j})$ gives a two-point based random gradient estimate with direction $\mathbf{u}_{i,j}$ and smoothing parameter $\mu > 0$. We remark that the random direction vectors in (3) can also be drawn from the standard Gaussian distribution (Nesterov & Spokoiny, 2015). However, the uniform distribution could be more useful in practice since it is defined in a bounded space rather than the whole real space required for Gaussian.

We highlight that unlike the first-order stochastic gradient estimate, the ZO gradient estimate (3) is a biased approximation to the true gradient of $f$. Instead, it becomes unbiased to the gradient of the randomized smoothing function $f_\mu$ (Duchi et al., 2012; Gao et al., 2014),

$$f_\mu(\mathbf{x}) = \mathbb{E}_{\mathbf{v}}[f(\mathbf{x} + \mu \mathbf{v})] = \frac{1}{n} \sum_{i=1}^{n} \mathbb{E}_{\mathbf{v}}[f_i(\mathbf{x} + \mu \mathbf{v})] = \frac{1}{n} \sum_{i=1}^{n} f_{i,\mu}(\mathbf{x}), \tag{4}$$

where $f_{i,\mu}$ gives the randomized smoothing version of $f_i$, and the random variable $\mathbf{v}$ follows a uniform distribution over the unit Euclidean ball. Clearly, there exists a gap between a ZO gradient estimate and the true gradient of $f$, but as will be evident later, such a gap can be measured through the smoothing function $f_\mu$.

**Motivations of ZO-signSGD.** Compared to SGD-type methods, the fast empirical convergence of signSGD and ZO-signSGD has been shown in the application of generating white-box and black-box adversarial examples (Goodfellow et al., 2015; Madry et al., 2018; Ilyas et al., 2018a). As mentioned in (Bernstein et al., 2018), the sign operation could mitigate the negative effect of (coordinate-wise) gradient noise of large variance. Recall that the ZO gradient estimate is a biased approximation to the true gradient, and thus, could suffer from having larger noise variance than (first-order) stochastic gradients. In this context, one could benefit from ZO-signSGD due to its robustness to gradient noise. In Appendix 1, we provide two concrete examples (Fig. A1 and Fig. A2) to confirm the aforementioned analysis. In Fig. A1, we show the robustness of ZO-signSGD against sparse noise perturbation through a toy quadratic optimization problem, originally introduced in (Bernstein et al., 2018) to motivate the fast convergence of signSGD against SGD. In Fig. A2, we show that gradient estimation via ZO oracle indeed encounters gradient noise of large variance. Thus, taking the sign of a gradient estimate might scale down the extremely noisy components.

**ZO-signSGD & technical challenges beyond signSGD** Algorithm 1 becomes ZO-signSGD as the ZO gradient estimate (3) is applied. We note that the extension from first order to ZO is nontrivial, as the proposed ZO-signSGD algorithm yields three key differences to signSGD.

First, ZO-signSGD has milder assumption on the choice of mini-batch sampling. Recall that signSGD in (Bernstein et al., 2018) achieves $O(1/\sqrt{T})$ convergence rate given the condition that the mini-batch size is sufficiently large, $b = O(T)$. However, this condition only becomes true when the mini-batch sample is randomly selected from $[n]$ *with replacement*, which is unusual when $n \leq T$. Here $[n]$ represents the integer set $\{1, 2, \ldots, n\}$. And signSGD fails to cover signGD when $b = n$, since sampling with replacement leads to $\mathcal{I}_k \neq [n]$ even if $b = n$. In the proposed ZO-signSGD algorithm, we will relax the assumption on mini-batch sampling.

Second, in ZO-signSGD both the ZO gradient estimator and the sign operator give rise to approximation errors to the true gradient. Although the statistical properties of ZO gradient estimates can be acquired with the aid of the randomized smoothing function (4), the use of mini-batch sampling without replacement introduces extra difficulty to bound the variance of ZO gradient estimates since mini-batch samples are no longer independent. Moreover, the sign-based descent algorithm evaluates the convergence error in the $\ell_1$-norm geometry, leading to a mismatch with the $\ell_2$-norm based gradient variance. Besides translating the the gradient norm from $\ell_1$ to $\ell_2$, the probabilistic convergence method (Ghadimi & Lan, 2013) is used to bound the eventual convergence error of ZO-signSGD.

Finally, beyond the standard ZO gradient estimator (3), we will cover multiple variants of ZO-signSGD for centralized or distributed optimization.

## 4 CONVERGENCE ANALYSIS OF ZO-SIGNSGD

In this section, we begin by stating assumptions used in our analysis. We then derive the convergence rate of ZO-signSGD for nonconvex optimization. Assumptions of problem (1) are listed as follows.

**A1:** Functions $\{f_i\}$ have $L$-Lipschitz continuous gradients, where $L \in (0, \infty)$.

**A2:** At time $k$, the gradient of $f_i$ is upper bounded by $\|\nabla f_i(\mathbf{x}_k)\|_2 \leq \sigma$ for $i \in [n]$.

Both A1 and A2 are the standard assumptions used in nonconvex optimization literature (Bernstein et al., 2018; Reddi et al., 2018; Chen et al., 2018). A1 implies the $L$-smoothness of $f_i$, namely, for any $\mathbf{x}$ and $\mathbf{y}$ we obtain $f_i(\mathbf{x}) - f_i(\mathbf{y}) \leq \langle \nabla f_i(\mathbf{y}), \mathbf{x} - \mathbf{y} \rangle + (L/2)\|\mathbf{x} - \mathbf{y}\|_i^2$. A2 implies the bounded variance of $\nabla f_i$ in (Bernstein et al., 2018, Assumption 3), namely, $\frac{1}{n} \sum_{i=1}^{n} \|\nabla f_i(\mathbf{x}) - \nabla f(\mathbf{x})\|_2^2 \leq 4\sigma^2$, where we have used the fact that $\|\nabla f(\mathbf{x})\|_2 \leq \sigma$ under A2. Throughout the paper, we assume that problem (1) is solvable, namely, $f(\mathbf{x}^*) > -\infty$ where $x^*$ is an optimal solution.

We recall that Algorithm 1 becomes ZO-signSGD when the gradient estimation step (3) is applied. For nonconvex problems, the convergence of an algorithm is typically measured by stationarity, e.g., using $\|\nabla f(\mathbf{x})\|_2^2$ in SGD (Ghadimi & Lan, 2013) and $\|\nabla f(\mathbf{x})\|_1$ in signSGD (Bernstein et al., 2018). For the latter, the $\ell_1$ geometry is met when quantifying the stochasticity through the (non-linear) sign operation. Different from signSGD, ZO-signSGD only obtains a biased estimate to the true gradient. In Proposition 1, we bypass such a bias by leveraging the randomized smoothing technique used for ZO optimization (Gao et al., 2014; Nesterov & Spokoiny, 2015; Duchi et al., 2015).

**Proposition 1** *Under A1, the outputs $\{\mathbf{x}_k\}_{k=0}^{T-1}$ of ZO-signSGD, i.e., Algorithm 1 with (3), satisfies*

$$\sum_{k=0}^{T-1} (\delta_k \mathbb{E}[\|\nabla f_\mu(\mathbf{x}_k)\|_1]) \leq \mathbb{E}[f_\mu(\mathbf{x}_0) - f_\mu(\mathbf{x}_T)] + \sum_{k=0}^{T-1} \left[ 2\delta_k \sqrt{d} \sqrt{\mathbb{E}[\|\hat{\mathbf{g}}_k - \nabla f_\mu(\mathbf{x}_k)\|_2^2]} \right] + \frac{dL}{2} \sum_{k=0}^{T-1} \delta_k^2,$$
(5)

*where the expectation is taken with respect to all the randomness of ZO-signSGD, $f_\mu$ is the randomized smoothing function of $f$ in (4), and $\hat{\mathbf{g}}_k = \text{GradEstimate}(\mathbf{x}_k)$ in (3).*

**Proof:** See Appendix 2. □

In Proposition 1, the rationale behind introducing the smoothing function $f_\mu$ is that $\nabla f_\mu(\mathbf{x}_k)$ is the mean of ZO gradient estimate $\hat{\mathbf{g}}_k$. And thus, the convergence of ZO-signSGD is now linked with the variance of $\hat{\mathbf{g}}_k$, i.e., $\mathbb{E}[\|\hat{\mathbf{g}}_k - \nabla f_\mu(\mathbf{x}_k)\|_2^2]$. This crucial relationship presented in Proposition 1 holds for a general class of signSGD-type algorithms that use different ZO gradient estimators. Spurred by (5), we next investigate the second-order moment of $\hat{\mathbf{g}}_k$ in Proposition 2.

**Proposition 2** *Under A1 and A2, the variance of ZO gradient estimate $\hat{\mathbf{g}}_k$ is upper bounded by*

$$\mathbb{E}\left[\|\hat{\mathbf{g}}_k - \nabla f_\mu(\mathbf{x}_k)\|_2^2\right] \leq \frac{4\alpha_b(q+1)}{bq}\sigma^2 + \frac{(2\alpha_b + \beta_b)}{bq}C(d,\mu),$$
(6)

*where $C(d,\mu) := 2d\sigma^2 + \mu^2 L^2 d^2/2$. In (6), $\alpha_b$ and $\beta_b$ are Boolean variables depending on the choice of mini-batch sampling,*

$$\begin{cases} \alpha_b = 1, \beta_b = 0 & \text{for mini-batch with replacement} \\ \alpha_b = I(b < n), \beta_b = I(b > 1) & \text{for mini-batch without replacement,} \end{cases}$$
(7)

*where $I(x > a)$ is the indicator function of $x$ with respect to the constraint $x > a$, and $I(x > a) = 1$ if $x > a$ and $0$ otherwise.*

**Proof:** See Appendix 3. □

Compared to the variance bound $(\sigma^2/b)$ of the stochastic gradient estimate of $f$ in signSGD (Bernstein et al., 2018), Proposition 2 provides a general result for the ZO gradient estimate $\hat{\mathbf{g}}_k$. It is clear that the bound in (6) contains two parts: $h_1 := \frac{4\alpha_b(q+1)}{bq}\sigma^2$ and $h_2 := \frac{(2\alpha_b + \beta_b)}{bq}C(d,\mu)$, where the former $h_1 = O(\sigma^2/b)$ characterizes the reduced variance (using $b$ mini-batch samples) for the stochastic gradient estimate of the smoothing function $f_\mu$, and the latter $h_2 = O(C(d,\mu)/(bq))$ reveals the dimension-dependent variance induced by ZO gradient estimate using $b$ mini-batch samples and $q$ random directions. If a stochastic gradient estimate of $f$ is used in signSGD, then $h_2$ is eliminated and the variance bound in (6) is reduced to $(\sigma^2/b)$.

Furthermore, Proposition 2 covers mini-batch sampling with and without replacement, while signSGD only considers the former case. For the latter case, Proposition 2 implies that if $b = n$ (i.e., $\mathcal{I}_k = [n]$ for ZO-signGD), then the variance $\mathbb{E}\left[\|\hat{\mathbf{g}}_k - \nabla f_\mu(\mathbf{x}_k)\|_2^2\right]$ is reduced to $O(C(d,\mu)/(nq))$, corresponding to $\alpha_b = 0$ and $\beta_b = 1$ in (7). In the other extreme case of $b = 1$, both the studied mini-batch schemes become identical, corresponding to $\alpha_b = 1$ and $\beta_b = 0$. Proposition 2 also implies that the use of large $b$ and $q$ reduces the variance of the gradient estimate, and will further improve the convergence rate.

With the aid of Proposition 1 and 2, we can then show the convergence rate of ZO-signSGD in terms of stationarity of the original function $f$. The remaining difficulty is how to bound the gap between $f$ and its smoothed version $f_\mu$. It has been shown in (Gao et al., 2014; Nesterov & Spokoiny, 2015) that there exists a tight relationship between $f_\mu$ and $f$ given the fact that the former is a convolution of the latter and the density function of a random perturbation $\mathbf{v}$ in (4). We demonstrate the convergence rate of ZO-signSGD in Theorem 1.

**Theorem 1** *Under A1 and A2, if we randomly pick $\mathbf{x}_R$ from $\{\mathbf{x}_k\}_{k=0}^{T-1}$ with probability $P(R = k) = \frac{\delta_k}{\sum_{k=0}^{T-1} \delta_k}$, then the convergence rate of ZO-signSGD is given by*

$$\mathbb{E}[\|\nabla f(\mathbf{x}_R)\|_2] \leq \frac{\sqrt{2}(f(\mathbf{x}_0) - f^* + \mu^2 L)}{\sum_{k=0}^{T-1} \delta_k} + \frac{dL}{\sqrt{2}} \frac{\sum_{k=0}^{T-1} \delta_k^2}{\sum_{k=0}^{T-1} \delta_k} + \frac{\mu L d}{\sqrt{2}}$$
$$+ \frac{2\sqrt{2}\sqrt{d}\sqrt{4\alpha_b(q+1)\sigma^2 + C(d,\mu)(2\alpha_b + \beta_b)}}{\sqrt{bq}},$$
(8)

*where $f^*$ denotes the minimum value.*

**Proof:** See Appendix 4. □

In Theorem 1, we translate the gradient norm from $\ell_1$ to $\ell_2$, and adopt a probabilistic output $\mathbf{x}_R$ (Ghadimi & Lan, 2013; Lei et al., 2017) to avoid exhaustive search over $\{\mathbf{x}_k\}$ for $\min_k \|\nabla f(\mathbf{x}_k)\|_2$. Note that the convergence rate of ZO-signSGD relies on the learning rate $\delta_k$, the problem size $d$, the smoothing parameter $\mu$, the mini-batch size $b$, and the number of random perturbations $q$ for ZO gradient estimation. We next obtain explicit dependence on these parameters by specifying Theorem 1.

If $\delta_k = \delta = O(\frac{1}{\sqrt{dT}})$ and $\mu = O(\frac{1}{\sqrt{dT}})$, then the convergence in (8) simplifies to

$$\mathbb{E}\left[\|\nabla f(\mathbf{x}_R)\|_2\right] \leq O\left(\frac{\sqrt{d}}{\sqrt{T}} + \frac{\sqrt{d}\sqrt{\alpha_b q + (\alpha_b + \beta_b)d}}{\sqrt{bq}}\right), \tag{9}$$

where $\alpha_b$ and $\beta_b$ were defined in (7), and $1 \leq (\alpha_b + \beta_b) \leq 2$. We provide several key insights on the convergence rate of ZO-signSGD through (9).

First, the convergence rate of ZO-signSGD is measured through $\|\nabla f(\mathbf{x}_R)\|_2$ rather than its *squared* counterpart $\|\nabla f(\mathbf{x}_R)\|_2^2$, where the latter was used in measuring the convergence of ZO-SGD. We recall from (Ghadimi & Lan, 2013, Theorem 3.2 & Corollary 3.3) that ZO-SGD yields the convergence error $\mathbb{E}\left[\|\nabla f(\mathbf{x}_R)\|_2^2\right] \leq O(\frac{\sqrt{d}}{\sqrt{T}})$. Since $\|\nabla f(\mathbf{x}_R)\|_2^2 \leq \|\nabla f(\mathbf{x}_R)\|_2$ as $\|\nabla f(\mathbf{x}_R)\|_2^2 \leq 1$, the convergence of ZO-signSGD meets a stricter criterion than that of ZO-SGD. The possible downside of ZO-signSGD is that it suffers an additional error of order $O(\frac{\sqrt{d}}{\sqrt{b}} + \frac{d}{\sqrt{bq}})$ in the worst case. The aforementioned results imply that ZO-signSGD could only converge to a neighborhood of a stationary point but with a fast convergence speed. Here the size of the neighborhood is controlled by the mini-batch size $b$ and the number of random direction vectors $q$.

Also, our convergence analysis applies to mini-batch sampling both with and without replacement. When $b \in [1, n)$, ZO-signSGD achieves $O(\frac{\sqrt{d}}{\sqrt{T}} + \frac{\sqrt{d}}{\sqrt{b}} + \frac{d}{\sqrt{bq}})$ convergence rate regardless of the choice of mini-batch sampling. When $b = n$, it is known from (9) that the use of mini-batch without replacement recovers ZO-signGD, yielding the convergence rate $O(\frac{\sqrt{d}}{\sqrt{T}} + \frac{d}{\sqrt{nq}})$. By contrast, the use of mini-batch with replacement leads to the worse convergence rate $O(\frac{\sqrt{d}}{\sqrt{T}} + \frac{d}{\sqrt{nq}} + \frac{\sqrt{d}}{\sqrt{n}})$. Clearly, as $b = n$ and $n < T$, ZO-signSGD using mini-batch with replacement fails to achieve the rate $O(\frac{\sqrt{d}}{\sqrt{T}})$ regardless of the choice of $q$. By contrast, ZO-signSGD using mini-batch without replacement recovers $O(\frac{\sqrt{d}}{\sqrt{T}})$ as $q = O(\frac{dT}{n})$. When $b > n$, ZO-signSGD is restricted to using mini-batch sampling with replacement. Similar to signSGD (Bernstein et al., 2018), we can obtain $O(\frac{\sqrt{d}}{\sqrt{T}})$ convergence rate as $b = O(T)$ and $q = O(\frac{dT}{n})$, where the dependence on $q$ is induced by the use of ZO gradient estimation.

## 5 VARIANTS OF ZO-SIGNSGD

Here we study three variants of ZO-signSGD, where the gradient will be estimated using a) the *central* difference of function values, b) the sign of ZO gradient estimates with *majority vote*, or c) the sign of ZO gradient estimates with majority vote for *distributed* optimization. That is,

$$a) \ \text{GradEstimate}(\mathbf{x}) = \frac{1}{bq} \sum_{i \in \mathcal{I}_k} \sum_{j=1}^{q} \frac{d[f_i(\mathbf{x} + \mu \mathbf{u}_{i,j}) - f_i(\mathbf{x} - \mu \mathbf{u}_{i,j})]\mathbf{u}_{i,j}}{2\mu} \tag{10}$$

$$b) \ \text{GradEstimate}(\mathbf{x}) = \frac{1}{bq} \sum_{i \in \mathcal{I}_k} \sum_{j=1}^{q} \text{sign}\left(\hat{\nabla} f_i(\mathbf{x}; \mathbf{u}_{i,j})\right), \tag{11}$$

$$c) \ \text{GradEstimate}(\mathbf{x}) = \frac{1}{M} \sum_{m=1}^{M} \text{sign}\left(\frac{1}{b_m q} \sum_{i \in \mathcal{I}_{m,k}} \sum_{j=1}^{q} \hat{\nabla} f_i(\mathbf{x}; \mathbf{u}_{i,j})\right), \tag{12}$$

where $\{\mathbf{u}_{i,j}\}$ and $\hat{\nabla} f_i(\mathbf{x}; \mathbf{u}_{i,j})$ have been defined in (3). The gradient estimator (12) is proposed for distributed optimization over a star network that consists of $M$ agents and 1 central processor. Each agent $m \in [M]$ has only access to $n_m$ data (in terms of $n_m$ individual costs $\{f_i\}$) with $\sum_{m=1}^{M} n_m = n$, and the mini-batch $\mathcal{I}_{m,k}$ per agent satisfies $b_m = |\mathcal{I}_{m,k}| \in [1, n_m]$. According to (12), the

central processor receives 1-bit compressed gradients $\text{sign}\left(\frac{1}{b_m q} \sum_{i \in \mathcal{I}_{m,k}} \sum_{j=1}^{q} \hat{\nabla} f_i(\mathbf{x}; \mathbf{u}_{i,j})\right)$ from $M$ agents and then performs the sign-based descent (2) and sends its 1-bit update back to every agent.

The ZO gradient estimator (10) was used in (Shamir, 2017) for bandit convex optimization and in (Ilyas et al., 2018a) for designing black-box adversarial attacks. Compared to the form of forward difference (3), the central difference (10) requires $b(q-1)$ times more function queries in gradient estimation. At the cost of more function queries, one may wonder if the convergence rate of ZO-signSGD can be further improved.

**Corollary 1** *Suppose that the conditions in Theorem 1 hold, ZO-signSGD with gradient estimator (10) yields the same convergence rate of ZO-signSGD that uses the estimator (3).*

**Proof:** Recall that Proposition 1 is independent of specific forms of gradient estimators, and thus holds for (10). Although Proposition 2 relies on the second-order moments of each gradient estimator, we prove that under A1 and A2, both (3) and (10) maintain the same statistical properties. As a result, Proposition 2 and Theorem 1 also hold for (10); see more details in Appendix 5. □

We next study the gradient estimator (11), whose sign is equivalent to the majority vote (i.e., the element-wise median) of signs of individual gradient estimates $\{\hat{\nabla} f_i(\mathbf{x}; \mathbf{u}_{i,j})\}$. It was shown in (Bernstein et al., 2018) that signSGD with majority vote has a better convergence rate under additional assumptions of unimodal symmetric noise distribution of coordinate-wise gradient estimates. In Corollary 2, we show that such a speed-up in convergence can also be achieved by ZO-signSGD with majority vote, which we refer to as 'ZO-M-signSGD'.

**Corollary 2** *Suppose that the conditions in Theorem 1 hold, and the distribution of gradient noise is unimodal and symmetric. Then, ZO-M-signSGD with $\delta_k = O(\frac{1}{\sqrt{dT}})$ and $\mu = O(\frac{1}{\sqrt{dT}})$ yields*

$$\mathbb{E}\left[\|\nabla f(\mathbf{x}_R)\|_2\right] = O\left(\sqrt{d}/\sqrt{T} + d/\sqrt{bq}\right). \tag{13}$$

**Proof:** See Appendix 6. □

We recall from Theorem 1 that under the same parameter setting of Corollary 2, ZO-signSGD yields $O(\frac{\sqrt{d}}{\sqrt{T}} + \frac{\sqrt{d}}{\sqrt{b}} + \frac{d}{\sqrt{bq}})$ convergence rate in the worst case. It is clear from (13) that the error correction term of order $\frac{\sqrt{d}}{\sqrt{b}}$ is eliminated in ZO-M-signSGD. Such an improvement in convergence is achieved under the condition of unimodal symmetric gradient noise. We remark that different from the stochastic gradient noise studied in (Bernstein et al., 2018), the ZO gradient estimation noise could violate this assumption. For example, in a scalar case, if the gradient estimate $g$ follows the distribution where $g = 1$ with probability 0.9, $g = -10$ with probability 0.1, then $\mathbb{E}[g] < 0$ and $\text{sign}(\mathbb{E}[g]) < 0$. However, $\mathbb{E}[\text{sign}(g)] > 0$. This implies that without the assumption of symmetry, the sign of gradient estimates with majority vote ($\mathbb{E}[\text{sign}(g)]$) can be in the opposite direction of the sign of averaged gradients ($\text{sign}(\mathbb{E}[g])$). Our results in the next section show that ZO-M-signSGD may not outperform ZO-signSGD.

Lastly, we focus on the gradient estimator (12), whose sign can be interpreted as the major vote of $M$ distributed agents about the sign of the true gradient (Bernstein et al., 2018). The resulting variant of ZO-signSGD is called 'ZO-D-signSGD', and its convergence rate is illustrated in Corollary 3. Compared to ZO-M-signSGD for centralized optimization, ZO-D-signSGD suffers an extra error correction term $O(\frac{\sqrt{d}}{\sqrt{n}})$ in the distributed setting. It is also worth mentioning that if $M = n$ and $q = 1$, then the gradient estimator (12) reduces to (11) with $\mathcal{I}_k = [n]$. In this case, Corollary 2 and 3 reach a consensus on $O(\frac{\sqrt{d}}{\sqrt{T}} + \frac{d}{\sqrt{n}})$ convergence error.

**Corollary 3** *Suppose that the conditions in Corollary 2 hold. ZO-M-signSGD with $b_m = \lfloor \frac{n}{M} \rfloor$, $\delta_k = O(\frac{1}{\sqrt{dT}})$ and $\mu = O(\frac{1}{\sqrt{dT}})$ yields*

$$\mathbb{E}\left[\|\nabla f(\mathbf{x}_R)\|_2\right] = O\left(\sqrt{d}/\sqrt{T} + \sqrt{d}/\sqrt{n} + d/\sqrt{nq}\right). \tag{14}$$

**Proof:** See Appendix 7. □

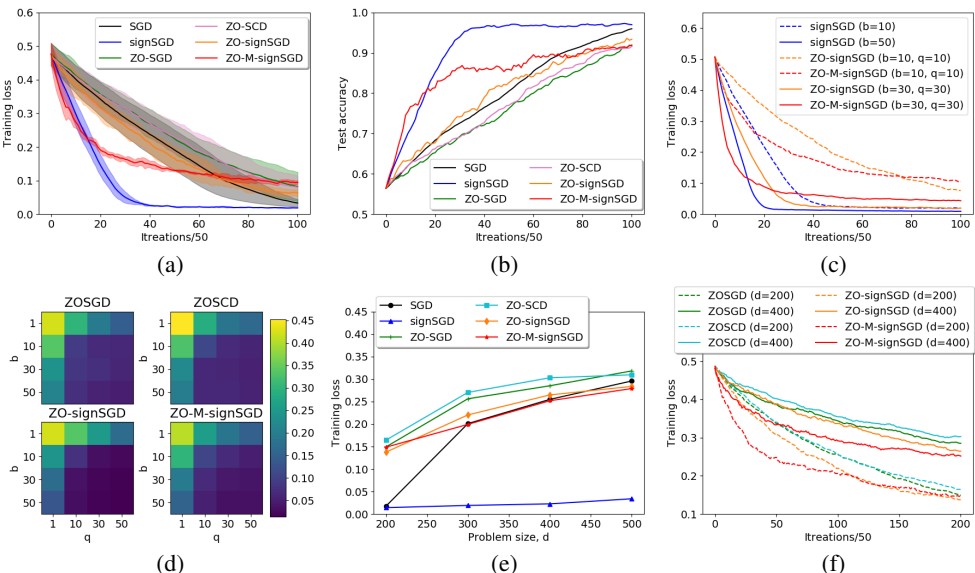

**Figure 1:** Performance comparison of ZO-signSGD, ZO-M-signSGD, ZO-SGD, ZO-SCD, signSGD and SGD under a synthetic dataset. The solid line represents the loss/accuracy averaged over 10 independent trials with random initialization, and the shaded region indicates the standard deviation of results over random trials. (a)-(b): Training loss and test accuracy versus iterations. (c)-(d): Effects of mini-batch size $q$ and number of random direction vectors $q$ on the convergence of studied algorithms. Here (c) presents the training loss versus iterations, and (d) is the heat map of the final loss for different values of $b$ and $q$. (e)-(f): Effects of problem size $d$. Here (e) shows the final training loss versus $d$, and (f) presents the convergence trajectory when $d \in \{200, 400\}$.

## 6 EXPERIMENTS

In this section, we empirically show the effectiveness of ZO-signSGD, and validate its convergence behavior on both synthetic and real-world datasets such as MNIST and CIFAR-10. For the synthetic experiment, we study the problem of binary classification in the least squared formulation. For the real-world application, we design adversarial examples from black-box neural networks as mentioned in Sec. 2. Throughout this section, we compare ZO-signSGD and its variants with SGD, signSGD (Bernstein et al., 2018), ZO-SGD (Ghadimi & Lan, 2013), and ZO-SCD (Lian et al., 2016).

**Binary classification** We consider the least squared problem with a *nonconvex* loss function (Xu et al., 2017; Liu et al., 2018b) $\min_{\mathbf{x} \in \mathbb{R}^d} \frac{1}{n} \sum_{i=1}^n (y_i - 1/(1 + e^{-\mathbf{a}_i^T \mathbf{x}}))^2$, which satisfies Assumption A2 by letting $\sigma = \max_i \{2\|\mathbf{a}_i\|_2\}$. Here instead of using the conventional cost function of logistic regression (a convex function), the considered least squared formulation is introduced to align with our nonconvex theoretical analysis. For generating the synthetic dataset, we randomly draw samples $\{\mathbf{a}_i\}$ from $\mathcal{N}(\mathbf{0}, \mathbf{I})$, and obtain the label $y_i = 1$ if $1/(1 + e^{-\mathbf{a}_i^T \mathbf{x}}) > 0.5$ and 0 otherwise. The number of training samples $\{\mathbf{a}_i, y_i\}$ is set by $n = 2000$ against 200 testing samples. We find the best constant learning rate for algorithms via a greedy search over $\eta \in [0.001, 0.1]$ (see Appendix 8.1 for more details), and we choose the smoothing parameter $\mu = 10/\sqrt{Td}$. Unless specified otherwise, let $b = q = 10$, $T = 5000$ and $d = 100$.

In Fig. 1, we report the training loss, the test accuracy, as well as the effects of algorithmic parameters on the convergence of the studied algorithms. We observe from Fig. 1-(a) and (b) that ZO-signSGD outperforms other ZO algorithms, and signSGD yields the best convergence performance once the first-order information is available. In Fig. 1-(c) and (d), we observe that the convergence performance of ZO algorithms is improved as $b$ and $q$ increase. In particular, ZO-signSGD and ZO-M-signSGD at $b = q = 30$ approach to the best result provided by signSGD. In Fig. 1-(e) and (f), the convergence of all algorithms degrades as the problem size $d$ increases. However, ZO-signSGD and ZO-M-signSGD converge faster than ZO-SGD and ZO-SCD. In Fig. 2, we demonstrate the convergence trajectory of different variants of ZO-signSGD for $b \in \{40, 400\}$. To make a fair comparison between ZO-signSGD and ZO-D-signSGD, let each of $M = 40$ agents use a mini-batch of size $b/M$. As we can see, ZO-signSGD outperforms ZO-M-signSGD and ZO-D-signSGD. And the convergence is improved as the mini-batch size increases. However, we observe that in all examples, ZO-signSGD

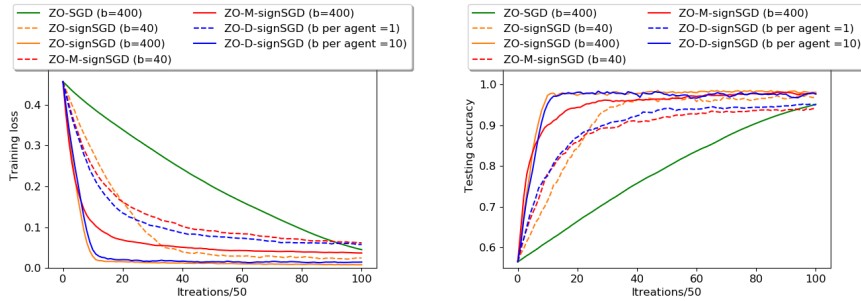

**Figure 2:** Training loss (left) and testing accuracy (right) of ZO-signSGD, ZO-M-signSGD, ZO-D-signSGD and ZO-SGD versus iterations.

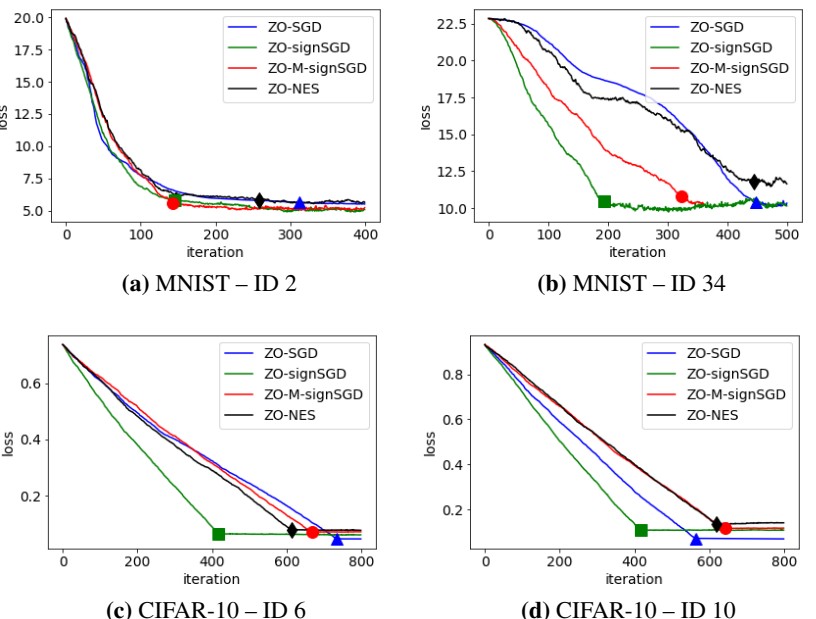

**(a)** MNIST – ID 2

**(b)** MNIST – ID 34

**(c)** CIFAR-10 – ID 6

**(d)** CIFAR-10 – ID 10

**Figure 3:** Black-box attacking loss versus iterations. The solid marker indicates the iteration number that finds the first successful adversarial example, and its loss corresponds to the squared $\ell_2$ distortion.

and its variants converge to moderate accuracy much faster than ZO-SGD, only within a few tens of iterations.

**Generating black-box adversarial examples**    Here we study adversarial robustness by generating adversarial examples from a black-box image classifier trained by a deep neural network (DNN) model; see details on problem formulation in Appendix 8.2. We recall from Sec. 2 that the task of black-box adversarial attack falls within the category of ZO optimization as one can only access to the input-output relation of the DNN while crafting adversarial examples.

The DNN models trained on MNIST and CIFAR-10 (Carlini & Wagner, 2017) are performed as the zeroth-order oracle[2]. We select one image from each class of MNIST and CIFAR-10 and separately implement black-box attacks using the same attacking loss function (see Appendix 8.2) but with different ZO optimization algorithms (ZO-SGD, ZO-signSGD and ZO-M-signSGD). We also set the same parameters for each method, i.e., $\mu = 0.01$, $q = 9$, and $\delta = 0.05$ for MNIST and $\delta = 0.0005$ for CIFAR-10, to accommodate to the dimension factor $d$. Moreover, we benchmark their performance with the natural evolution strategy (NES) based two-point gradient estimator in (Ilyas et al., 2018a) for solving the same attacking loss function, where the sign of gradient estimate is also used in the

---

[2] https://github.com/carlini/nn_robust_attacks

**Table 1:** Iteration comparison of attacking black-box DNN on MNIST (image ID 2).

| Iteration | 0 | 40 | 80 | 120 | 160 | 200 | 240 | 280 | 312 | 356 |
|---|---|---|---|---|---|---|---|---|---|---|
| ZO-SGD | | | | | | | | | | |
| Classified as | 1 | 1 | 1 | 1 | 1 | 1 | 1 | 1 | 4 | 4 |
| Iteration | 0 | 40 | 80 | 120 | 145 | 202 | 240 | 280 | 320 | 359 |
| ZO-signSGD | | | | | | | | | | |
| Classified as | 1 | 1 | 1 | 1 | 4 | 4 | 4 | 4 | 4 | 4 |
| Iteration | 0 | 40 | 80 | 120 | 142 | 200 | 240 | 279 | 320 | 360 |
| ZO-M-signSGD | | | | | | | | | | |
| Classified as | 1 | 1 | 1 | 1 | 4 | 4 | 4 | 4 | 4 | 4 |
| Iteration | 0 | 40 | 80 | 120 | 160 | 200 | 258 | 287 | 321 | 359 |
| ZO-NES | | | | | | | | | | |
| Classified as | 1 | 1 | 1 | 1 | 1 | 1 | 4 | 4 | 4 | 4 |

descent step. We call the resulting black-box attack generation method 'ZO-NES'. Similar to (10), NES computes the ZO gradient estimate using the central difference of two function values. Thus, one iteration of ZO-NES requires $2q$ function queries and thus we set $q = 5$ to align with the number of function queries used in other ZO methods. All methods use the the same natural image as the initial point for finding adversarial examples.

Fig. 3 shows the plots of black-box attacking loss versus iterations (more results are shown in Appendix 8.3). We find that ZO-signSGD usually takes significantly less iterations than other methods to find the first successful adversarial example with a similar attacking loss. For MNIST, the average iteration over all attacked images in Table A1 to find the first successful adversarial example is 184 for ZO-SGD, 103 for ZO-signSGD, 151 for ZO-M-signSGD, and 227 for ZO-NES. Their corresponding average $\ell_2$ distortion is 2.345 for ZO-SGD, 2.381 for ZO-signSGD, 2.418 for ZO-M-signSGD, and 2.488 for ZO-NES. For CIFAR-10, the average iteration over all attacked images in Table A2 to find the first successful adversarial example is 302 for ZO-SGD, 250 for ZO-signSGD, 389 for ZO-M-signSGD, and 363 for ZO-NES. Their corresponding average $\ell_2$ distortion is 0.177 for ZO-SGD, 0.208 for ZO-signSGD, 0.219 for ZO-M-signSGD, and 0.235 for ZO-NES. As a visual illustration, we compare the adversarial examples of a hand-written digit "1" of each attacking method at different iterations in Table 1, corresponding to Fig. 3-(a). As we can see, ZO-signSGD and ZO-M-signSGD can reduce roughly 54% of iterations (around 600 less model queries) than ZO-SGD to find the first successful adversarial example. Given the first successful adversarial example, we observe that ZO-signSGD yields slightly higher $\ell_2$ distortion than ZO-SGD. This is not surprising since Theorem 1 suggests that ZO-signSGD might not converge to a solution of very high accuracy but it can converge to moderate accuracy sufficient for black-box attacks at a very fast speed. Note that the first successful adversarial examples generated by different ZO methods are all visually similar to the original ones but lead to different top-1 predictions; see more results in Appendix 8.3. In addition, we observe that ZO-NES is not as effective as ZO-signSGD in either query efficiency (given by the number of iterations to achieve the first successful attack) or attack distortion. Thus, compared to ZO-NES, ZO-signSGD offers a provable and an efficient black-box adversarial attacking method.

## 7 CONCLUSION

Motivated by the impressive convergence behavior of (first-order) signSGD and the empirical success in crafting adversarial examples from black-box ML models, in this paper we rigorously prove the $O(\sqrt{d}/\sqrt{T})$ convergence rate of ZO-signSGD and its variants under mild conditions. Compared to signSGD, ZO-signSGD suffers a slowdown (proportional to the problem size $d$) in convergence rate, however, it enjoys the gradient-free advantages. Compared to other ZO algorithms, we corroborate the superior performance of ZO-signSGD on both synthetic and real-word datasets, particularly for its application to black-box adversarial attacks. In the future, we would like to generalize our analysis to nonsmooth and nonconvex constrained optimization problems.

ACKNOWLEDGMENTS

This work was supported by the MIT-IBM Watson AI Lab. Mingyi Hong and Xiangyi Chen are supported partly by an NSF grant CMMI-1727757,and by an AFOSR grant 15RT0767.

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

# APPENDIX

## 1 MOTIVATING EXAMPLES OF FAST CONVERGENCE OF ZO-SIGNSGD

### 1.1 EXAMPLE OF SPARSE NOISE PERTURBATION

We consider to minimize the function $f(\mathbf{x}) = \frac{1}{2}\|\mathbf{x}\|_2^2$. Similar to (Bernstein et al., 2018, Figure A.1), we assume that the ZO gradient estimate of $f(\mathbf{x})$ and its first-order gradient $\nabla f(\mathbf{x}) = \mathbf{x}$ suffer from a sparse noise vector $\mathbf{v}$, where $v_1 \in \mathcal{N}(0, 100^2)$, and $v_i = 0$ for $i \geq 2$. As a result, the used descent direction at iteration $t$ is given by $\hat{\nabla} f(\mathbf{x}_t) + \mathbf{v}$ or $\nabla f(\mathbf{x}_t) + \mathbf{v}$. Fig. A1 presents the convergence performance of 5 algorithms: SGD, signSGD, ZO-SGD, ZO-signSGD and its variant using the central difference based gradient estimator (10). Here we tune a constant learning rate finding 0.001 best for SGD and ZO-SGD and 0.01 best for signSGD and its ZO variants. As we can see, sign-based first-order and ZO algorithms converge much faster than the stochastic gradient-based descent algorithms. This is not surprising since the presence of extremely noisy component $v_1$ leads to an inaccurate gradient value, and thus degrades the convergence of SGD and ZO-SGD. By contrast, the sign information is more robust to outliers and thus leads to better convergence performance of sign SGD and its variants. We also note that the convergence trajectory of ZO-signSGD using the gradient estimator (10) coincides with that using the gradient estimator (3) given by the forward difference of two function values.

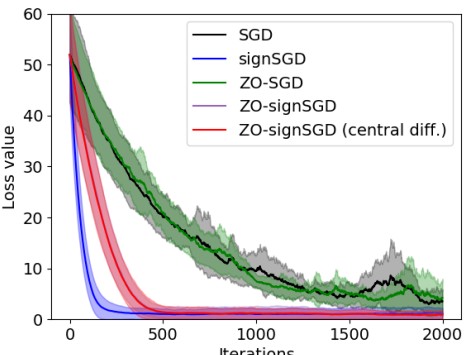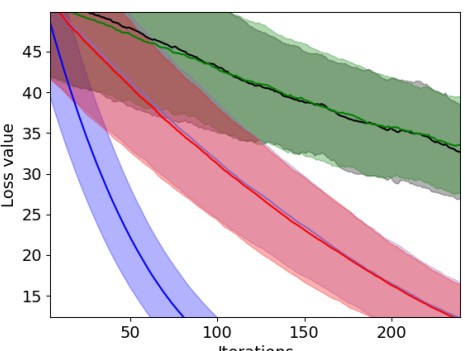

**Figure A1:** Comparison of different gradient-based and gradient sign-based first-order and ZO algorithms in the example of sparse noise perturbation. The solid line represents the loss averaged over 10 independent trials with random initialization, and the shaded region indicates the standard deviation of results over random trials. Left: Loss value against iterations for SGD, signSGD, ZO-SGD, ZO-signSGD and ZO-signSGD using the central difference based gradient estimator (10). Right: Local regions to highlight the effect of the gradient estimators (3) and (10) on the convergence of ZO-signSGD.

### 1.2 STATISTICS OF GRADIENT ESTIMATES

The intuition behind why ZO-signSGD could outperform ZO-SGD is that the sign operation can mitigate the negative effect of (coordinate-wise) gradient noise of large variance. To confirm this point, we examine the coordinate-wise variance of gradient noises during an entire training run of the binary classifier provided in the first experiment of Sec. 6. At each iteration, we perform an additional 100 random trials to obtain the statistics of gradient estimates. In Fig. A2-(a), we present the $\ell_1$ norm of the mean of gradient estimates (over 100 trials) versus the number of iterations. As we can see, both signSGD and ZO-signSGD outperform SGD and ZO-SGD, evidenced by a fast decrease of the $\ell_1$ norm of gradient estimate. In Fig. A2-(b), we present the coordinate-wise gradient noise variance (over 100 trails at each coordinate) against the number of iterations. It is not surprising that compared to first-order methods, ZO methods suffer gradient noise of larger variance. In this scenario, we could benefit from ZO-signSGD since taking the sign of gradient estimates might scale down extremely noisy components. Indeed, we observe a significant decrease of the noise variance while performing ZO-signSGD compared to ZO-SGD.

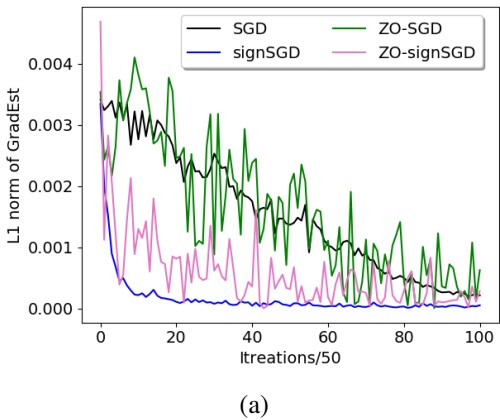
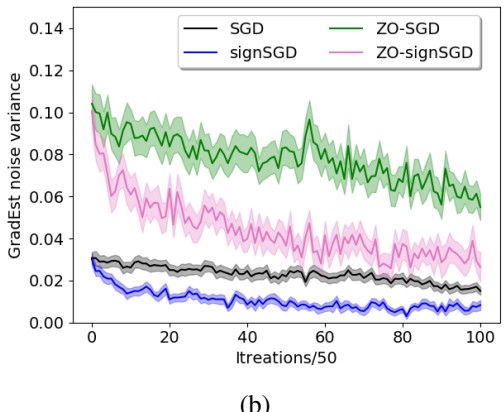

$$(a) \qquad\qquad\qquad (b)$$

**Figure A2:** Statistics of gradient estimates during an entire training run of the binary classifier provided in the first experiment of Sec. 6. a) The $\ell_1$ norm of the mean of gradient estimates versus iteration. b) Coordinate-wise gradient noise variance versus iteration. The solid line represents the variance averaged over all coordinates, and the shaded region indicates the corresponding standard deviation with respect to all coordinates at each iteration.

## 2 PROOF OF PROPOSITION 1

Based on the definition of the smoothing function $f_\mu$, for any $\mathbf{x}$ and $\mathbf{y}$ we have

$$
\begin{aligned}
\|\nabla f_\mu(x) - \nabla f_\mu(y)\|_2 =& \|\mathbb{E}_\mathbf{v}[\nabla_\mathbf{x} f(\mathbf{x} + \mu\mathbf{v}) - \nabla_\mathbf{y} f(\mathbf{y} + \mu\mathbf{v})]\|_2 \\
\leq& \mathbb{E}[\|\nabla_\mathbf{x} f(\mathbf{x} + \mu\mathbf{v}) - \nabla_\mathbf{y} f(\mathbf{y} + \mu\mathbf{v})\|_2] \leq L\|\mathbf{x} - \mathbf{y}\|_2,
\end{aligned} \tag{15}
$$

where the first inequality holds due to Jensen's inequality, and the second inequality holds due to A1. It is known from (15) that $f_\mu$ has $L$-Lipschitz continuous gradient.

By the $L$-smoothness of $f_\mu$, we obtain that

$$
\begin{aligned}
f_\mu(\mathbf{x}_{k+1}) \leq& f_\mu(\mathbf{x}_k) + \langle \nabla f_\mu(\mathbf{x}_k), \mathbf{x}_{k+1} - \mathbf{x}_k \rangle + \frac{L}{2}\|\mathbf{x}_{k+1} - \mathbf{x}_k\|_2^2 \\
\overset{(2)}{=}& f_\mu(\mathbf{x}_k) - \delta_k \langle \nabla f_\mu(\mathbf{x}_k), \mathrm{sign}(\hat{\mathbf{g}}_k) \rangle + \frac{L}{2}\delta_k^2\|\mathrm{sign}(\hat{\mathbf{g}}_k)\|_2^2 \\
=& f_\mu(\mathbf{x}_k) - \delta_k\|\nabla f_\mu(\mathbf{x}_k)\|_1 + \frac{\delta_k^2 dL}{2} \\
& + 2\delta_k \sum_{i=1}^d |(\nabla f_\mu(\mathbf{x}_k))_i| \mathcal{I}\left[\mathrm{sign}(\hat{\mathbf{g}}_{k,i}) \neq \mathrm{sign}((\nabla f_\mu(\mathbf{x}_k))_i)\right],
\end{aligned} \tag{16}
$$

where $(\nabla f_\mu(\mathbf{x}))_i$ denotes the $i$th element of $\nabla f_\mu(\mathbf{x})$.

Taking expectation for both sides of (16), we obtain that

$$
\begin{aligned}
\mathbb{E}[f_\mu(\mathbf{x}_{k+1}) - f_\mu(\mathbf{x}_k)] \leq& -\delta_k\|\nabla f_\mu(\mathbf{x}_k)\|_1 + \frac{\delta_k^2 dL}{2} \\
& + 2\delta_k \sum_{i=1}^d |(\nabla f_\mu(\mathbf{x}_k))_i| \mathrm{Prob}\left[\mathrm{sign}(\hat{\mathbf{g}}_{k,i}) \neq \mathrm{sign}((\nabla f_\mu(\mathbf{x}_k))_i)\right].
\end{aligned} \tag{17}
$$

Similar to (Bernstein et al., 2018, Theorem 1), we relax $\mathrm{Prob}\left[\mathrm{sign}(\hat{\mathbf{g}}_{k,i}) \neq \mathrm{sign}((\nabla f_\mu(\mathbf{x}_k))_i)\right]$ by Markov's inequality,

$$
\begin{aligned}
\mathrm{Prob}\left[\mathrm{sign}(\hat{\mathbf{g}}_{k,i}) \neq \mathrm{sign}((\nabla f_\mu(\mathbf{x}_k))_i)\right] \leq& \mathrm{Prob}[|\hat{\mathbf{g}}_{k,i} - (\nabla f_\mu(\mathbf{x}_k))_i| \geq |(\nabla f_\mu(\mathbf{x}_k))_i|] \\
\leq& \frac{\mathbb{E}[|\hat{\mathbf{g}}_{k,i} - (\nabla f_\mu(\mathbf{x}_k))_i|]}{|(\nabla f_\mu(\mathbf{x}_k))_i|}.
\end{aligned} \tag{18}
$$

Substituting (18) into (17), we obtain

$$\mathbb{E}[f_\mu(\mathbf{x}_{k+1}) - f_\mu(\mathbf{x}_k)] \leq -\delta_k \|\nabla f_\mu(\mathbf{x}_k)\|_1 + \frac{\delta_k^2 dL}{2} + 2\delta_k \sum_{i=1}^{d} \mathbb{E}[|\hat{\mathbf{g}}_{k,i} - (\nabla f_\mu(\mathbf{x}_k))_i|]$$

$$= -\delta_k \|\nabla f_\mu(\mathbf{x}_k)\|_1 + \frac{\delta_k^2 dL}{2} + 2\delta_k \mathbb{E}[\|\hat{\mathbf{g}}_k - \nabla f_\mu(\mathbf{x}_k)\|_1]$$

$$\leq -\delta_k \|\nabla f_\mu(\mathbf{x}_k)\|_1 + \frac{\delta_k^2 dL}{2} + 2\delta_k \sqrt{d} \mathbb{E}[\|\hat{\mathbf{g}}_k - \nabla f_\mu(\mathbf{x}_k)\|_2]$$

$$\leq -\delta_k \|\nabla f_\mu(\mathbf{x}_k)\|_1 + \frac{\delta_k^2 dL}{2} + 2\delta_k \sqrt{d}\sqrt{\mathbb{E}[\|\hat{\mathbf{g}}_k - \nabla f_\mu(\mathbf{x}_k)\|_2^2]}, \qquad (19)$$

where the second inequality holds due to $\|\mathbf{x}\|_1 \leq \sqrt{d}\|\mathbf{x}\|_2$, and the last inequality holds by applying Jensen's inequality to the concave function $\sqrt{\cdot}$, i.e., $\mathbb{E}[\|\hat{\mathbf{g}}_k - \nabla f_\mu(\mathbf{x}_k)\|_2] \leq \sqrt{\mathbb{E}[\|\hat{\mathbf{g}}_k - \nabla f_\mu(\mathbf{x}_k)\|_2^2]}$.

Taking sum of both sides of (19), we then obtain (5). $\qquad \square$

## 3  PROOF OF PROPOSITION 2

We recall from (3) that

$$\hat{\mathbf{g}}_k = \frac{1}{b} \sum_{i \in \mathcal{I}_k} \hat{\nabla} f_i(\mathbf{x}_k), \ \hat{\nabla} f_i(\mathbf{x}_k) = \frac{1}{q} \sum_{j=1}^{q} \hat{\nabla} f_i(\mathbf{x}_k; \mathbf{u}_{i,j}). \qquad (20)$$

Let $\mathbf{z}_i := \hat{\nabla} f_i(\mathbf{x}_k) - \nabla f_\mu(\mathbf{x}_k)$ and $\mathbf{z}_{i,j} = \hat{\nabla} f_i(\mathbf{x}_k; \mathbf{u}_{i,j}) - \nabla f_\mu(\mathbf{x}_k)$. Thus,

$$\hat{\mathbf{g}}_k - \nabla f_\mu(\mathbf{x}_k) = \frac{1}{b} \sum_{i \in \mathcal{I}_k} \mathbf{z}_i = \frac{1}{bq} \sum_{i \in \mathcal{I}_k} \sum_{j=1}^{q} \mathbf{z}_{i,j} \qquad (21)$$

where there are two sources of randomness: a) minibatch sampling $i \in \mathcal{I}_k$, and b) the random direction sampling $\mathbf{u} = \mathbf{u}_{i,j}$. Note that these two sources of randomness are independent, and the random direction samples $\{\mathbf{u}_{i,j}\}$ are i.i.d..

Next, we discuss two types of mini-batch sampling: a) mini-batch samples without replacement, and b) mini-batch samples with replacement.

Suppose that $\mathcal{I}_k$ is a uniform random subset of $[n]$ (no replacement), motivated by (Lei et al., 2017, Lemma A.1) we introduce a new variable $W_i = I(i \in \mathcal{I}_k)$, where $I$ is an indicator function, and $I(i \in \mathcal{I}_k) = 1$ if $i \in \mathcal{I}_k$, and 0 otherwise. As a result, we have

$$\mathbb{E}[W_i^2] = \mathbb{E}[W_i] = \frac{b}{n}, \quad \mathbb{E}[W_i W_j] = \frac{b(b-1)}{n(n-1)}, \ i \neq j. \qquad (22)$$

From (21), the variance of $\hat{\mathbf{g}}_k$ is given by

$$\mathbb{E}\left[\left\|\frac{1}{b}\sum_{i \in \mathcal{I}_k} \mathbf{z}_i\right\|_2^2\right] = \frac{1}{b^2}\left(\sum_{i=1}^{n}\left(\mathbb{E}_{i \in \mathcal{I}_k}[W_i^2]\mathbb{E}_\mathbf{u}[\|\mathbf{z}_i\|_2^2]\right) + \sum_{i \neq j}\mathbb{E}_{i,j \in \mathcal{I}_k}[W_i W_j]\langle \mathbb{E}_\mathbf{u}[\mathbf{z}_i], \mathbb{E}_\mathbf{u}[\mathbf{z}_j]\rangle\right)$$

$$= \frac{1}{b^2}\left(\frac{b}{n}\sum_{i=1}^{n}\mathbb{E}_\mathbf{u}[\|\mathbf{z}_i\|_2^2] + \frac{b(b-1)}{n(n-1)}\left\|\sum_{i=1}^{n}\mathbb{E}_\mathbf{u}[\mathbf{z}_i]\right\|_2^2 - \frac{b(b-1)}{n(n-1)}\sum_{i=1}^{n}\|\mathbb{E}_\mathbf{u}[\mathbf{z}_i]\|_2^2\right)$$

$$\overset{(a)}{=} \frac{1}{bn}\sum_{i=1}^{n}\mathbb{E}_\mathbf{u}[\|\mathbf{z}_i\|_2^2] - \frac{b-1}{b(n-1)n}\sum_{i=1}^{n}\|\mathbb{E}_\mathbf{u}[\mathbf{z}_i]\|_2^2$$

$$= \frac{1}{bn}\sum_{i=1}^{n}\mathbb{E}_\mathbf{u}[\|\mathbf{z}_i\|_2^2] - \frac{b-1}{b(n-1)n}\sum_{i=1}^{n}\mathbb{E}_\mathbf{u}[\|\mathbf{z}_i\|_2^2] + \frac{b-1}{b(n-1)n}\sum_{i=1}^{n}\left(\mathbb{E}_\mathbf{u}[\|\mathbf{z}_i\|_2^2] - \|\mathbb{E}_\mathbf{u}[\mathbf{z}_i]\|_2^2\right)$$

$$\overset{(b)}{=} \frac{n-b}{(n-1)}\frac{1}{bn}\sum_{i=1}^{n}\mathbb{E}_\mathbf{u}[\|\mathbf{z}_i\|_2^2] + \frac{b-1}{b(n-1)n}\sum_{i=1}^{n}\mathbb{E}_\mathbf{u}[\|\hat{\nabla} f_i(\mathbf{x}_k) - \nabla f_{i,\mu}(\mathbf{x}_k)\|_2^2]. \qquad (23)$$

In (23), the equality (a) holds since $\mathbb{E}_\mathbf{u}[\mathbf{z}_i] = \nabla f_{i,\mu}(\mathbf{x}_k) - \nabla f_\mu(\mathbf{x}_k)$ and $f_\mu(\mathbf{x}) = \frac{1}{n}\sum_{i=1}^{n} f_{i,\mu}(\mathbf{x})$ from (4), where we have used the fact that $\mathbb{E}_\mathbf{u}[\hat{\nabla} f_i(\mathbf{x}_k)] = \hat{\nabla} f_{i,\mu}(\mathbf{x}_k)$ (Liu et al., 2018c, Lemma. 1), and recall that $f_{i,\mu}$ denotes the smoothing function of $f_i$. The above implies that $\sum_{i=1}^{n}\mathbb{E}_\mathbf{u}[\mathbf{z}_i] = \sum_i \nabla f_{i,\mu}(\mathbf{x}_k) - n\nabla f_\mu(\mathbf{x}_k) = \mathbf{0}$. And the equality (b) holds due to $\mathbb{E}_\mathbf{u}[\|\mathbf{z}_i\|_2^2] - \|\mathbb{E}_\mathbf{u}[\mathbf{z}_i]\|_2^2 = \mathbb{E}_\mathbf{u}\|\mathbf{z}_i - \mathbb{E}_\mathbf{u}[\mathbf{z}_i]\|_2^2$.

On the other hand, suppose that the mini-batch $\mathcal{I}_k$ contains i.i.d. samples (namely, with replacement), the vectors $\{\mathbf{z}_i\}$ are then i.i.d. under both mini-batch sampling and random direction sampling. Therefore, we obtain that

$$
\mathbb{E}\left[\left\|\frac{1}{b}\sum_{i\in\mathcal{I}_k}\mathbf{z}_i\right\|_2^2\right] = \frac{1}{b^2}\left[\mathbb{E}\left[\sum_{i\in\mathcal{I}_k}\|\mathbf{z}_i\|_2^2\right] + \mathbb{E}\left[\sum_{i\neq j, i,j\in\mathcal{I}_k}\langle\mathbf{z}_i,\mathbf{z}_j\rangle\right]\right]
$$

$$
= \frac{1}{b}\mathbb{E}_{\mathbf{u}}[\mathbb{E}_{i\in\mathcal{I}_k}[\|\mathbf{z}_i\|_2^2]] = \frac{1}{bn}\sum_{i=1}^n\mathbb{E}_{\mathbf{u}}[\|\mathbf{z}_i\|_2^2], \tag{24}
$$

where the second equality holds since $\mathbb{E}_{i\in\mathcal{I}_k,\mathbf{u}}[\mathbf{z}_i] = \frac{1}{n}\sum_{i=1}^n\mathbb{E}_{\mathbf{u}}[\mathbf{z}_i] = \frac{1}{n}\sum_{i=1}^n\nabla f_{i,\mu}(\mathbf{x}_k) - \nabla f_\mu(\mathbf{x}_k) = \mathbf{0}$.

Combining both (23) and (24), we obtain that

$$
\mathbb{E}\left[\left\|\frac{1}{b}\sum_{i\in\mathcal{I}_k}\mathbf{z}_i\right\|_2^2\right] \leq \frac{\alpha_b}{bn}\sum_{i=1}^n\mathbb{E}_{\mathbf{u}}[\|\mathbf{z}_i\|_2^2] + \frac{\beta_b}{bn}\sum_{i=1}^n\mathbb{E}_{\mathbf{u}}[\|\hat{\nabla}f_i(\mathbf{x}_k) - \nabla f_{i,\mu}(\mathbf{x}_k)\|_2^2]. \tag{25}
$$

In (25), $\alpha_b = 1$ and $\beta_b = 0$ if the mini-batch contains i.i.d. samples from $[n]$ with replacement, and $\alpha_b = I(b < n)$ and $\beta_b = I(b > 1)$ if samples are randomly selected without replacement.

In (25), we next bound $\frac{1}{n}\sum_i\mathbb{E}_{\mathbf{u}}[\|\mathbf{z}_i\|_2^2]$,

$$
\frac{1}{n}\sum_{i=1}^n\mathbb{E}_{\mathbf{u}}[\|\mathbf{z}_i\|_2^2] \overset{(21)}{=} \frac{1}{n}\sum_{i=1}^n\mathbb{E}_{\mathbf{u}}\left\|\frac{1}{q}\sum_{j=1}^q\mathbf{z}_{i,j}\right\|_2^2 = \frac{1}{nq^2}\sum_i\left[\sum_j\mathbb{E}_{\mathbf{u}}[\|\mathbf{z}_{i,j}\|_2^2] + \sum_{j\neq k}\langle\mathbb{E}_{\mathbf{u}}[\mathbf{z}_{i,j}],\mathbb{E}_{\mathbf{u}}[\mathbf{z}_{i,k}]\rangle\right]
$$

$$
= \frac{1}{nq^2}\sum_i\left[q\mathbb{E}_{\mathbf{u}}[\|\mathbf{z}_{i,1}\|_2^2] + (q^2-q)\|\mathbb{E}_{\mathbf{u}}[\mathbf{z}_{i,1}]\|_2^2\right]
$$

$$
= \frac{1}{qn}\sum_i\mathbb{E}_{\mathbf{u}}[\|\mathbf{z}_{i,1}\|_2^2] + \frac{q-1}{qn}\sum_i\|\mathbb{E}_{\mathbf{u}}[\mathbf{z}_{i,1}]\|_2^2, \tag{26}
$$

where we have used the facts that $\mathbb{E}_{\mathbf{u}}[\mathbf{z}_{i,j}] = \mathbb{E}_{\mathbf{u}}[\mathbf{z}_{i,1}]$ and $\mathbb{E}_{\mathbf{u}}[\|\mathbf{z}_{i,j}\|_2^2] = \mathbb{E}_{\mathbf{u}}[\|\mathbf{z}_{i,1}\|_2^2]$ for any $j$ since random direction vectors $\{\mathbf{u}_{i,j}\}_{j=1}^q$ are i.i.d. samples.

In (26), we further bound $\sum_i\mathbb{E}_{\mathbf{u}}[\|\mathbf{z}_{i,1}\|_2^2]$,

$$
\frac{1}{n}\sum_i\mathbb{E}_{\mathbf{u}}[\|\mathbf{z}_{i,1}\|_2^2] = \frac{1}{n}\sum_i\mathbb{E}_{\mathbf{u}}\|\hat{\nabla}f_i(\mathbf{x}_k;\mathbf{u}_{i,1}) - \nabla f_\mu(\mathbf{x}_k)\|_2^2
$$

$$
= \frac{1}{n}\sum_i\mathbb{E}_{\mathbf{u}}[\|\hat{\nabla}f_i - \nabla f_{i,\mu} + \nabla f_{i,\mu} - \nabla f_\mu\|_2^2]
$$

$$
\leq \frac{2}{n}\sum_i\mathbb{E}_{\mathbf{u}}[\|\hat{\nabla}f_i - \nabla f_{i,\mu}\|_2^2] + \frac{2}{n}\sum_i\|\nabla f_{i,\mu} - \nabla f_\mu\|_2^2, \tag{27}
$$

where for ease of notation, let $\hat{\nabla}f_i := \hat{\nabla}f_i(\mathbf{x}_k;\mathbf{u}_{i,1})$, $\hat{\nabla}f_{i,\mu} := \hat{\nabla}f_{i,\mu}(\mathbf{x}_k)$ and $\nabla f_\mu := \nabla f_\mu(\mathbf{x}_k)$. According to (Liu et al., 2018c, Lemma 1), the first term at RHS of (27) yields

$$
\mathbb{E}_{\mathbf{u}}[\|\hat{\nabla}f_i - \nabla f_{i,\mu}\|_2^2] \leq 2d\|\nabla f_i\|_2^2 + \frac{\mu^2L^2d^2}{2} \leq 2d\sigma^2 + \frac{\mu^2L^2d^2}{2} := C(d,\mu), \tag{28}
$$

where the last inequality holds due to A2. Based on the definition of $f_\mu$, the second term at RHS of (27) yields

$$
\frac{1}{n}\sum_i\|\nabla f_{i,\mu} - \nabla f_\mu\|_2^2 = \frac{1}{n}\sum_i\|\mathbb{E}_{\mathbf{v}}[\nabla f_i(\mathbf{x}_k+\mu\mathbf{v}) - \nabla f(\mathbf{x}_k+\mu\mathbf{v})]\|_2^2
$$

$$
\leq \frac{1}{n}\sum_i\mathbb{E}_{\mathbf{v}}[\|\nabla f_i(\mathbf{x}_k+\mu\mathbf{v}) - \nabla f(\mathbf{x}_k+\mu\mathbf{v})\|_2^2] \leq 4\sigma^2, \tag{29}
$$

where we have used the Jensen's inequality and $\frac{1}{n}\sum_{i=1}^n\|\nabla f_i(\mathbf{x}) - \nabla f(\mathbf{x})\|_2^2 \leq 4\sigma^2$ under A2.

Substituting (28) and (29) into (27), we have

$$
\frac{1}{n}\sum_i\mathbb{E}_{\mathbf{u}}[\|\mathbf{z}_{i,1}\|_2^2] = \frac{1}{n}\sum_i\mathbb{E}_{\mathbf{u}}[\|\hat{\nabla}f_i - \nabla f_\mu\|_2^2] \leq 4d\sigma^2 + \mu^2L^2d^2 + 8\sigma^2 = 2C(d,\mu) + 8\sigma^2, \tag{30}
$$

where $C(d,\mu)$ was defined in (28).

We are now ready to bound (26). Based on $\frac{1}{n}\sum_i \|\mathbb{E}_{\mathbf{u}}[\mathbf{z}_{i,1}]\|_2^2 = \frac{1}{n}\sum_i \|\nabla f_{i,\mu} - \nabla f_\mu\|_2^2 \leq 4\sigma^2$ from (29), and substituting (30) into (26), we obtain that

$$\frac{1}{n}\sum_i \mathbb{E}_{\mathbf{u}}[\|\mathbf{z}_i\|_2^2] \leq \frac{2C(d,\mu) + 8\sigma^2}{q} + \frac{4(q-1)\sigma^2}{q}. \tag{31}$$

In (25), we also need to bound $\mathbb{E}_{\mathbf{u}}[\|\hat{\nabla} f_i(\mathbf{x}_k) - \nabla f_{i,\mu}(\mathbf{x}_k)\|_2^2]$

$$\mathbb{E}_{\mathbf{u}}\left[\left\|\hat{\nabla} f_i(\mathbf{x}_k) - \nabla f_{i,\mu}(\mathbf{x}_k)\right\|_2^2\right] \overset{(20)}{=} \mathbb{E}_{\mathbf{u}}\left[\left\|\frac{1}{q}\sum_{j=1}^q \left(\hat{\nabla} f_i(\mathbf{x}; \mathbf{u}_{i,j}) - \nabla f_{i,\mu}(\mathbf{x})\right)\right\|_2^2\right]$$

$$\overset{(a)}{=} \frac{1}{q}\mathbb{E}_{\mathbf{u}}\left[\left\|\hat{\nabla} f_i(\mathbf{x}; \mathbf{u}_{i,1}) - \nabla f_{i,\mu}(\mathbf{x})\right\|_2^2\right] \overset{(28)}{\leq} \frac{1}{q}\left(2d\sigma^2 + \frac{\mu^2 L^2 d^2}{2}\right) = \frac{C(d,\mu)}{q}, \tag{32}$$

where the equality (a) holds since $\mathbb{E}_{\mathbf{u}}[\hat{\nabla} f_i(\mathbf{x}; \mathbf{u}_{i,j})] = \nabla f_{i,\mu}(\mathbf{x})$ for any $j$, given by (Liu et al., 2018c, Lemma 1).

Substituting (31) and (32) into (25)

$$\mathbb{E}\left[\|\hat{\mathbf{g}}_k - \nabla f_\mu(\mathbf{x}_k)\|_2^2\right] = \mathbb{E}\left[\left\|\frac{1}{b}\sum_{i\in\mathcal{I}_k}\mathbf{z}_i\right\|_2^2\right] \leq \frac{\alpha_b}{b}\frac{2C(d,\mu) + 4\sigma^2 + 4\sigma^2 q}{q} + \frac{\beta_b C(d,\mu)}{bq}$$

$$= \frac{4\alpha_b(q+1)}{bq}\sigma^2 + \frac{C(d,\mu)}{bq}(2\alpha_b + \beta_b). \tag{33}$$

$\square$

# 4 PROOF OF THEOREM 1

Substituting (6) into (5), we obtain

$$\mathbb{E}\left[\sum_{k=0}^{T-1}\delta_k\|\nabla f_\mu(\mathbf{x}_k)\|_1\right] \leq \mathbb{E}[f_\mu(\mathbf{x}_0) - f_\mu(\mathbf{x}_T)] + \sum_{k=0}^{T-1}\left[2\delta_k\frac{\sqrt{d}}{\sqrt{bq}}\sqrt{4\alpha_b(q+1)\sigma^2 + C(d,\mu)(2\alpha_b + \beta_b)}\right]$$

$$+ \frac{dL}{2}\sum_{k=0}^{T-1}\delta_k^2. \tag{34}$$

It is known from (Liu et al., 2018c, Lemma 1) that

$$\|\nabla f(\mathbf{x})\|_2^2 \leq 2\|\nabla f_\mu(\mathbf{x})\|_2^2 + \frac{\mu^2 L^2 d^2}{2}, \quad |f_\mu(\mathbf{x}) - f(\mathbf{x})| \leq \frac{\mu^2 L}{2}. \tag{35}$$

From (35) we have $f_\mu(\mathbf{x}_0) - f(\mathbf{x}_0) \leq \frac{\mu^2 L}{2}$ and $f^* - f_\mu^* \leq \frac{\mu^2 L}{2}$, where $f_\mu^* = \min_{\mathbf{x}} f_\mu(\mathbf{x})$ and $f^* = \min_{\mathbf{x}} f(\mathbf{x})$. This yields $f_\mu(\mathbf{x}_0) - f(\mathbf{x}_0) + f^* - f_\mu^* \leq \mu^2 L$, and thus

$$f_\mu(\mathbf{x}_0) - f_\mu(\mathbf{x}_T) \leq f_\mu(\mathbf{x}_0) - f_\mu^* \leq (f(\mathbf{x}_0) - f^*) + \mu^2 L. \tag{36}$$

Substituting (36) into (34), we obtain

$$\mathbb{E}\left[\sum_{k=0}^{T-1}\delta_k\|\nabla f_\mu(\mathbf{x}_k)\|_1\right] \leq \mathbb{E}[f(\mathbf{x}_0) - f^*)] + \mu^2 L$$

$$+ \sum_{k=0}^{T-1}\left[2\delta_k\frac{\sqrt{d}}{\sqrt{bq}}\sqrt{4\alpha_b(q+1)\sigma^2 + C(d,\mu)(2\alpha_b + \beta_b)}\right] + \frac{dL}{2}\sum_{k=0}^{T-1}\delta_k^2. \tag{37}$$

Due to $\|\nabla f_\mu(\mathbf{x}_k)\|_2 \leq \|\nabla f_\mu(\mathbf{x}_k)\|_1$ and dividing $\sum_{k=0}^{T-1}\delta_k$ for both sides of (37), we obtain that

$$\mathbb{E}\left[\sum_{k=0}^{T-1}\frac{\delta_k}{\sum_{k=0}^{T-1}\delta_k}\|\nabla f_\mu(\mathbf{x}_k)\|_2\right] \leq \frac{f(\mathbf{x}_0) - f^* + \mu^2 L}{\sum_{k=0}^{T-1}\delta_k} + \frac{2\sqrt{d}}{\sqrt{bq}}\sqrt{4\alpha_b(q+1)\sigma^2 + C(d,\mu)(2\alpha_b + \beta_b)}$$

$$+ \frac{dL}{2}\frac{\sum_{k=0}^{T-1}\delta_k^2}{\sum_{k=0}^{T-1}\delta_k}. \tag{38}$$

By introducing the random variable $R$ with probability $P(R = k) = \frac{\delta_k}{\sum_{k=0}^{T-1} \delta_k}$, we then obtain that

$$\mathbb{E}\left[\|\nabla f_\mu(\mathbf{x}_R)\|_2\right] = \mathbb{E}\left[\mathbb{E}_R[\|\nabla f_\mu(\mathbf{x}_R)\|_2]\right] = \mathbb{E}\left[\sum_{k=0}^{T-1} P(R = k)\|\nabla f_\mu(\mathbf{x}_k)\|_2\right]. \tag{39}$$

Based on (35), we further have

$$\mathbb{E}\left[\|\nabla f(\mathbf{x}_R)\|_2\right] \leq \mathbb{E}\left[\sqrt{2\|\nabla f_\mu(\mathbf{x}_R)\|_2^2 + \frac{\mu^2 L^2 d^2}{2}}\right]$$

$$\leq \sqrt{2}\mathbb{E}[\|\nabla f_\mu(\mathbf{x}_R)\|_2] + \frac{\mu L d}{\sqrt{2}}, \tag{40}$$

where we have used the fact that $\sqrt{a^2 + b^2} \leq (a + b)$ for $a, b \geq 0$.

Substituting (38) and (39) into (40), we finally obtain (8). $\qquad\square$

## 5    PROOF OF COROLLARY 1

Upon defining $\tilde{\nabla} f_i(\mathbf{x}; \mathbf{u}) = \frac{d[f_i(\mathbf{x}+\mu\mathbf{u})-f_i(\mathbf{x}-\mu\mathbf{u})]\mathbf{u}}{2\mu}$ (against $\hat{\nabla} f_i(\mathbf{x}; \mathbf{u})$ in (3)), our major task is to derive the first- and second-order moments of $\tilde{\nabla} f_i(\mathbf{x}; \mathbf{u})$.

Given $\mathbf{x}$, we first study the mean of $\tilde{\nabla} f_i(\mathbf{x}; \mathbf{u})$,

$$\mathbb{E}_\mathbf{u}\left[\tilde{\nabla} f_i(\mathbf{x}; \mathbf{u})\right] = \mathbb{E}_\mathbf{u}\left[\frac{d}{2\mu}(f_i(\mathbf{x}+\mu\mathbf{u}) - f_i(\mathbf{x}-\mu\mathbf{u}))\mathbf{u}\right]$$

$$= \mathbb{E}_\mathbf{u}\left[\frac{d}{2\mu}f_i(\mathbf{x}+\mu\mathbf{u})\mathbf{u}\right] + \mathbb{E}_\mathbf{u}\left[\frac{d}{2\mu}f_i(\mathbf{x}-\mu\mathbf{u})(-\mathbf{u})\right] \overset{(a)}{=} \mathbb{E}_\mathbf{u}\left[\frac{d}{\mu}f_i(\mathbf{x}+\mu\mathbf{u})\mathbf{u}\right]$$

$$\overset{(b)}{=} \mathbb{E}_\mathbf{u}\left[\frac{d}{\mu}\left(f_i(\mathbf{x}+\mu\mathbf{u}) - f_i(\mathbf{x})\right)\mathbf{u}\right] = \mathbb{E}_\mathbf{u}\left[\hat{\nabla} f_i(\mathbf{x}; \mathbf{u})\right] \overset{(c)}{=} \nabla f_{i,\mu}(\mathbf{x}), \tag{41}$$

where (a) holds since the distribution of $\mathbf{u}$ is symmetric around the origin, (b) holds since $\mathbb{E}[\mathbf{u}] = \mathbf{0}$, and (c) holds since $\nabla f_{i,\mu}(\mathbf{x}) = \mathbb{E}_\mathbf{u}\left[\frac{d}{\mu}f_i(\mathbf{x}+\mu\mathbf{u})\mathbf{u}\right]$ obtained from (Gao et al., 2014, Lemma 4.1). It is clear from (41) that $\mathbb{E}_\mathbf{u}\left[\tilde{\nabla} f_i(\mathbf{x}; \mathbf{u})\right] = \mathbb{E}_\mathbf{u}\left[\hat{\nabla} f_i(\mathbf{x}; \mathbf{u})\right]$.

We next study the second-order moment of $\tilde{\nabla} f_i(\mathbf{x}; \mathbf{u})$,

$$\mathbb{E}_\mathbf{u}\left[\|\tilde{\nabla} f_i(\mathbf{x}; \mathbf{u})\|^2\right] = \frac{d^2}{4\mu^2}\mathbb{E}_\mathbf{u}\left[(f_i(\mathbf{x}+\mu\mathbf{u}) - f_i(\mathbf{x}-\mu\mathbf{u}))^2\|\mathbf{u}\|^2\right]$$

$$\leq \frac{d^2}{2\mu^2}\mathbb{E}_\mathbf{u}\left[(f_i(\mathbf{x}+\mu\mathbf{u}) - f_i(\mathbf{x}))^2 + (f_i(\mathbf{x}) - f_i(\mathbf{x}-\mu\mathbf{u}))^2\right]$$

$$= \mathbb{E}_\mathbf{u}\left[\left\|\frac{d}{\mu}(f_i(\mathbf{x}+\mu\mathbf{u}) - f_i(\mathbf{x}))\mathbf{u}\right\|^2\right] = \mathbb{E}_\mathbf{u}\left[\|\hat{\nabla} f_i(\mathbf{x}; \mathbf{u})\|^2\right], \tag{42}$$

where we have used the fact that $\|\mathbf{u}\|_2 = 1$.

Based on (41) and (42), we can conclude that both (3) and (10) maintain the same statistical properties. Following the proofs of Proposition 2 and Theorem 1, it is not difficult to reach the convergence rate in (8). $\qquad\square$

## 6    PROOF OF COROLLARY 2

Let $\hat{\mathbf{g}}_k^{i,j} := \hat{\nabla} f_i(\mathbf{x}_k; \mathbf{u}_{i,j})$. If we replace $\hat{\mathbf{g}}_k$ with $\hat{\mathbf{g}}_k^{i,j}$ in (18), we then have

$$|(\nabla f_\mu(\mathbf{x}_k))_l|\,\text{Prob}\left[\text{sign}(\hat{g}_{k,l}^{i,j}) \neq \text{sign}((\nabla f_\mu(\mathbf{x}_k))_l)\right] \leq \mathbb{E}[|\hat{g}_{k,l}^{i,j} - (\nabla f_\mu(\mathbf{x}_k))_l|], \tag{43}$$

where $\hat{g}_{k,l}^{i,j}$ is the $l$th coordinate of $\hat{\mathbf{g}}_k^{i,j}$.

By letting $b = 1$ and $q = 1$ in Proposition 2, we know that $\mathbb{E}\left[\|\hat{\mathbf{g}}_k^{i,j} - \nabla f_\mu(\mathbf{x}_k)\|_2^2\right]$ is upper bounded. Moreover, by Jensen's inequality we have

$$\mathbb{E}[|\hat{g}_{k,l}^{i,j} - (\nabla f_\mu(\mathbf{x}_k))_l|] \leq \sqrt{\mathbb{E}[(\hat{g}_{k,l}^{i,j} - (\nabla f_\mu(\mathbf{x}_k))_l)^2]} := \xi_l, \tag{44}$$

where $\xi_l$ is finite since $\mathbb{E}\left[\|\hat{\mathbf{g}}_k^{i,j} - \nabla f_\mu(\mathbf{x}_k)\|_2^2\right]$ is upper bounded.

Substituting (44) into (43), we have

$$|(\nabla f_\mu(\mathbf{x}_k))_l|\,\text{Prob}\left[\text{sign}(\hat{g}_{k,l}^{i,j}) \neq \text{sign}((\nabla f_\mu(\mathbf{x}_k))_l)\right] \leq \xi_l. \tag{45}$$

With the new gradient estimate $\bar{\mathbf{g}}_k = \sum_{i \in \mathcal{I}_k} \sum_{j=1}^q \text{sign}(\hat{\mathbf{g}}_k^{i,j})$ in (11), we require to bound

$$\text{Prob}\left[\text{sign}(\bar{g}_{k,l}) \neq \text{sign}((\nabla f_\mu(\mathbf{x}_k))_l)\right], \tag{46}$$

where $\bar{g}_{k,l}$ is the $l$th coordinate of $\bar{\mathbf{g}}_k$.

We recall that $\hat{g}_{k,l}^{i,j}$ is an unbiased stochastic approximation to gradient component $(\nabla f_\mu(\mathbf{x}_k))_l$ with variance $\xi_l^2$. Under the assumption that the noise distribution is unimodal and symmetric, we know from (Bernstein et al., 2018, Lemma D1) that

$$\text{Prob}\left[\text{sign}(\hat{g}_{k,l}^{i,j}) \neq \text{sign}((\nabla f_\mu(\mathbf{x}_k))_l)\right] := Q \leq \begin{cases} \frac{2}{9}\frac{1}{S^2} & S > \frac{2}{\sqrt{3}} \\ \frac{1}{2} - \frac{S}{2\sqrt{3}} & \text{otherwise} \end{cases} < \frac{1}{2}, \tag{47}$$

where $S := |(\nabla f_\mu(\mathbf{x}_k))_l|/\xi_l$.

Let $Z$ count the number of estimates $\{\hat{g}_{k,l}^{i,j}\}$ yielding the correct sign of $((\nabla f_\mu(\mathbf{x}_k))_l$. Thus, the probability of error in (46) is equal to

$$\text{Prob}\left[\text{sign}(\bar{g}_{k,l}) \neq \text{sign}((\nabla f_\mu(\mathbf{x}_k))_l)\right] = \text{Prob}\left[Z \leq \frac{bq}{2}\right]. \tag{48}$$

Following the proof of (Bernstein et al., 2018, Theorem 2b) under (47), it is not difficult to obtain that

$$\text{Prob}\left[Z \leq \frac{bq}{2}\right] \leq \frac{1}{\sqrt{bq}S}. \tag{49}$$

That is,

$$|(\nabla f_\mu(\mathbf{x}_k))_l|\,\text{Prob}\left[\text{sign}(\bar{g}_{k,l}) \neq \text{sign}((\nabla f_\mu(\mathbf{x}_k))_l)\right] \leq \frac{\xi_l}{\sqrt{bq}}. \tag{50}$$

Replace $\hat{\mathbf{g}}_k$ with $\bar{\mathbf{g}}_k$ in (17), we obtain that

$$\mathbb{E}[f_\mu(\mathbf{x}_{k+1}) - f_\mu(\mathbf{x}_k)] \leq -\delta_k\|\nabla f_\mu(\mathbf{x}_k)\|_1 + \frac{\delta_k^2 dL}{2}$$

$$+ 2\delta_k \sum_{l=1}^d |(\nabla f_\mu(\mathbf{x}_k))_l|\text{Prob}\left[\text{sign}(\bar{g}_{k,l}) \neq \text{sign}((\nabla f_\mu(\mathbf{x}_k))_l)\right]$$

$$\overset{(50)}{\leq} -\delta_k\|\nabla f_\mu(\mathbf{x}_k)\|_1 + \frac{\delta_k^2 dL}{2} + 2\delta_k \frac{1}{\sqrt{bq}}\|\boldsymbol{\xi}\|_1$$

$$\leq -\delta_k\|\nabla f_\mu(\mathbf{x}_k)\|_1 + \frac{\delta_k^2 dL}{2} + 2\delta_k \frac{\sqrt{d}}{\sqrt{bq}}\sqrt{\|\boldsymbol{\xi}\|_2^2}$$

$$\overset{(44)}{=} -\delta_k\|\nabla f_\mu(\mathbf{x}_k)\|_1 + \frac{\delta_k^2 dL}{2} + 2\delta_k \frac{\sqrt{d}}{\sqrt{bq}}\sqrt{\mathbb{E}[\|\hat{\mathbf{g}}_k^{i,j} - \nabla f_\mu(\mathbf{x}_k)\|_2^2]}. \tag{51}$$

Compared (51) with (19), the standard deviation $\sqrt{\mathbb{E}[\|\hat{\mathbf{g}}_k^{i,j} - \nabla f_\mu(\mathbf{x}_k)\|_2^2]}$ is reduced by the factor $1/\sqrt{bq}$. According to Proposition 2, let $b = q = 1$, we obtain

$$\mathbb{E}\left[\|\hat{\mathbf{g}}_k^{i,j} - \nabla f_\mu(\mathbf{x}_k)\|_2^2\right] \leq 8\alpha_b\sigma^2 + (2\alpha_b + \beta_b)C(d,\mu). \tag{52}$$

Based on (51)-(52) and following the proof of Theorem 1, we have

$$\mathbb{E}\left[\|\nabla f(\mathbf{x}_R)\|_2\right] \leq \sqrt{2}\frac{f(\mathbf{x}_0) - f^* + \mu^2 L}{\sum_{k=0}^{T-1} \delta_k}$$

$$+ \frac{2\sqrt{2}\sqrt{d}}{\sqrt{bq}}\sqrt{8\alpha_b\sigma^2 + C(d,\mu)(2\alpha_b + \beta_b)} + \frac{dL}{\sqrt{2}}\frac{\sum_{k=0}^{T-1}\delta_k^2}{\sum_{k=0}^{T-1}\delta_k} + \frac{\mu L d}{\sqrt{2}}, \tag{53}$$

where $C(d,\mu) := 2d\sigma^2 + \mu^2 L^2 d^2/2$.

If $\mu = O(\frac{1}{\sqrt{dT}})$ and $\delta_k = O(\frac{1}{\sqrt{dT}})$, then the convergence rate simplifies to $O(\frac{\sqrt{d}}{\sqrt{T}} + \frac{d}{\sqrt{bq}})$. $\qquad\square$

## 7    PROOF OF COROLLARY 3

Let $\hat{\mathbf{g}}_k^m := \frac{1}{b_m q} \sum_{i \in \mathcal{I}_{m,k}} \sum_{j=1}^q \hat{\nabla} f_i(\mathbf{x}_k; \mathbf{u}_{i,j})$ and $\bar{\mathbf{g}}_k = \frac{1}{M} \sum_{m=1}^M \mathrm{sign}(\hat{\mathbf{g}}_k^m)$. Following (43)-(50), we can similarly obtain that

$$|(\nabla f_\mu(\mathbf{x}_k))_l| \, \mathrm{Prob}\left[\mathrm{sign}(\bar{g}_{k,l}) \neq \mathrm{sign}((\nabla f_\mu(\mathbf{x}_k))_l)\right] \leq \frac{\xi_l}{\sqrt{M}}, \tag{54}$$

where $\xi_l := \sqrt{\mathbb{E}[(\hat{g}_{k,l}^m - (\nabla f_\mu(\mathbf{x}_k))_l)^2]}$. By mimicking the derivation in (51), we have

$$\mathbb{E}[f_\mu(\mathbf{x}_{k+1}) - f_\mu(\mathbf{x}_k)] \leq -\delta_k \|\nabla f_\mu(\mathbf{x}_k)\|_1 + \frac{\delta_k^2 dL}{2} + 2\delta_k \frac{\sqrt{d}}{\sqrt{M}} \sqrt{\mathbb{E}[\|\hat{\mathbf{g}}_k^m - \nabla f_\mu(\mathbf{x}_k)\|_2^2]}. \tag{55}$$

According to Proposition 2, let $b = \lfloor n/M \rfloor$, we obtain

$$\mathbb{E}\left[\|\hat{\mathbf{g}}_k^m - \nabla f_\mu(\mathbf{x}_k)\|_2^2\right] \leq \frac{4(q+1)}{bq}\sigma^2 + \frac{3}{bq}C(d,\mu). \tag{56}$$

Based on (55)-(56) and following the proof of Theorem 1, we have

$$\mathbb{E}\left[\|\nabla f(\mathbf{x}_R)\|_2\right] \leq \sqrt{2} \frac{f(\mathbf{x}_0) - f^* + \mu^2 L}{\sum_{k=0}^{T-1} \delta_k}$$
$$+ \frac{2\sqrt{2}\sqrt{d}}{\sqrt{M b q}} \sqrt{4(q+1)\sigma^2 + 3C(d,\mu)} + \frac{dL}{\sqrt{2}} \frac{\sum_{k=0}^{T-1} \delta_k^2}{\sum_{k=0}^{T-1} \delta_k} + \frac{\mu L d}{\sqrt{2}}, \tag{57}$$

where $C(d,\mu) := 2d\sigma^2 + \mu^2 L^2 d^2/2$.

Since $\mu = O(\frac{1}{\sqrt{dT}})$, $\delta_k = O(\frac{1}{\sqrt{dT}})$ and $bM \approx n$, then the convergence rate simplifies to $O(\frac{\sqrt{d}}{\sqrt{T}} + \frac{\sqrt{d}}{\sqrt{n}} + \frac{d}{\sqrt{nq}})$. $\square$

## 8    SUPPLEMENTAL EXPERIMENTS

### 8.1    SYNTHETIC EXPERIMENTS

In Fig. A3, we demonstrate the effect of the learning rate $\delta$ on the training loss of SGD, signSGD, ZO-SGD, ZO-SCD, ZO-signSGD and ZO-M-signSGD. We observe that compared to the gradient-based algorithms (SGD, ZO-SGD and ZO-SCD), the gradient sign-based algorithms support a more flexible choice of learning rates (corresponding to less variance), since the sign operation has an normalization effect to reduce oscillations in convergence. We find $\delta = 0.1$ best for SGD, $\delta = 0.009$ best for signSGD, $\delta = 0.1$ best for ZOSGD and ZOSCD, $\delta = 0.0178$ best for ZOsignSGD, and $\delta = 0.0501$ best for ZO-M-signSGD.

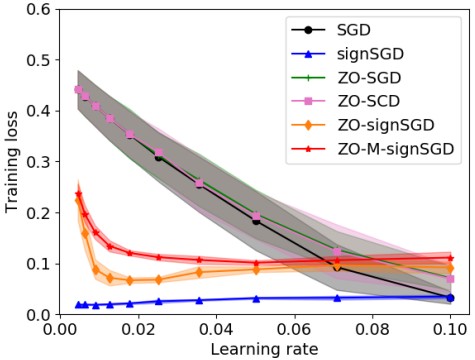

**Figure A3:** The training loss at the last iteration versus the constant learning rate $\delta \in [10^{-3}, 0.1]$. Here the solid line represents the loss averaged over 10 independent trials with random initialization, and the shaded region indicates the standard deviation of results over random trials.

## 8.2 BLACK-BOX ATTACK FORMULATION

The target DNN classifier $F = [F_1, F_2, \ldots, F_K]$ takes an image as an input and produces the classification predictions (here the probability distribution) of $K$ image classes, where $F_k$ denotes the prediction of class $k$. Given $F$, an adversarial example $\mathbf{x}$ of a legitimate example $\mathbf{x}_0$ means it is visually similar to $\mathbf{x}_0$ but will yield a different top-1 prediction than $\mathbf{x}_0$.

Let $(\mathbf{x}_0, t_0)$ denote a legitimate image $\mathbf{x}_0$ and its groundtruth label $t_0 \in \{1, 2, \ldots, K\}$. Without loss of generality, we assume the pixel value range of $\mathbf{x}_0$ lies within $[-0.5, 0.5]^d$. By using the $\tanh$ transformation on the adversarial image $\mathbf{x}$ such that $\mathbf{x} = \tanh(\mathbf{w})/2$, where $\mathbf{w} \in \mathbb{R}^d$, we adopt the untargeted black-box attacking loss designed in (Chen et al., 2017), which is defined as

$$\underset{\mathbf{w} \in \mathbb{R}^d}{\text{minimize}} \ c \cdot \max\{\log F_{t_0}(\tanh(\mathbf{w})/2) - \max_{j \neq t_0} \log F_j(\tanh(\mathbf{w})/2), 0\} + \|\tanh(\mathbf{w})/2 - \mathbf{x}_0\|_2^2, \quad (58)$$

where $c$ is a regularization coefficient and $\mathbf{x} = \tanh(\mathbf{w})/2$ ensures $\mathbf{x}$ lies within the valid image space $[-0.5, 0.5]^d$. The first term in the attacking objective is a hinge loss function that penalizes the top-1 prediction of $\mathbf{x}$ being $t_0$. The $\log F(\cdot)$ operation preserves the class prediction ranking and better handles numerical stability. The second term encourages the visual similarity between $\mathbf{x}$ and $\mathbf{x}_0$ through penalizing their squared $\ell_2$ difference (i.e., the squared distortion). In our experiment, we set $c = 1$ for MNIST and $c = 0.1$ for CIFAR-10. We also note that different from the use of signed gradients in existing black-box attacks (e.g., Ilyas et al. (2018a); Bhagoji et al. (2018)) due to the explicit $\ell_\infty$ perturbation constraint, the use and benefit of ZO-signSGD for solving (58) in our experiment are non-trivial since the attacking objective does not impose any $\ell_\infty$ constraint.

## 8.3 ADDITIONAL BLACK-BOX ATTACKING RESULTS ON MNIST AND CIFAR-10

**Table A1:** First successful adversarial examples attacking black-box DNN on MNIST.

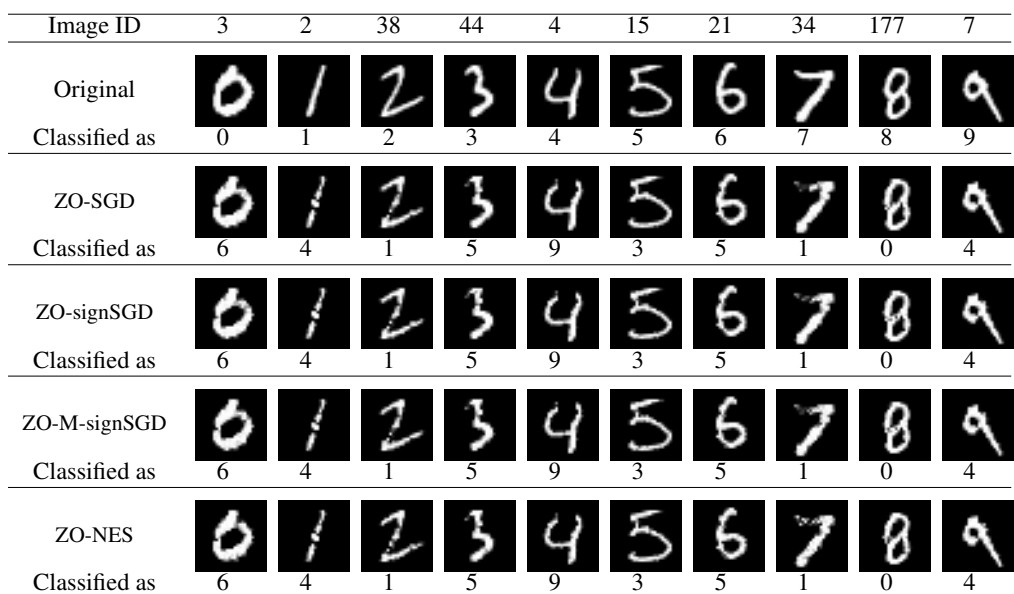

| Image ID | 3 | 2 | 38 | 44 | 4 | 15 | 21 | 34 | 177 | 7 |
|---|---|---|---|---|---|---|---|---|---|---|
| Original | | | | | | | | | | |
| Classified as | 0 | 1 | 2 | 3 | 4 | 5 | 6 | 7 | 8 | 9 |
| ZO-SGD | | | | | | | | | | |
| Classified as | 6 | 4 | 1 | 5 | 9 | 3 | 5 | 1 | 0 | 4 |
| ZO-signSGD | | | | | | | | | | |
| Classified as | 6 | 4 | 1 | 5 | 9 | 3 | 5 | 1 | 0 | 4 |
| ZO-M-signSGD | | | | | | | | | | |
| Classified as | 6 | 4 | 1 | 5 | 9 | 3 | 5 | 1 | 0 | 4 |
| ZO-NES | | | | | | | | | | |
| Classified as | 6 | 4 | 1 | 5 | 9 | 3 | 5 | 1 | 0 | 4 |

**Table A2:** First successful adversarial examples attacking black-box DNN on CIFAR-10.

| Image ID | 10 | 6 | 25 | 68 | 36 | 33 | 5 | 17 | 1 | 28 |
|---|---|---|---|---|---|---|---|---|---|---|
| Original | | | | | | | | | | |
| Classified as | airplane | automobile | bird | cat | deer | dog | frog | horse | ship | truck |
| ZO-SGD | | | | | | | | | | |
| Classified as | bird | truck | truck | dog | horse | cat | cat | dog | truck | automobile |
| ZO-signSGD | | | | | | | | | | |
| Classified as | bird | truck | truck | dog | horse | cat | cat | dog | truck | automobile |
| ZO-M-signSGD | | | | | | | | | | |
| Classified as | bird | truck | truck | dog | horse | cat | cat | dog | automobile | automobile |
| ZO-NES | | | | | | | | | | |
| Classified as | bird | truck | truck | dog | horse | cat | cat | dog | automobile | automobile |

**Table A3:** Iteration comparison of attacking black-box DNN on MNIST (image ID 177).

| Iteration | 0 | 30 | 60 | 90 | 120 | 162 | 181 | 210 | 241 | 271 |
|---|---|---|---|---|---|---|---|---|---|---|
| ZO-SGD | | | | | | | | | | |
| Classified as | 8 | 8 | 8 | 8 | 8 | 0 | 0 | 0 | 0 | 0 |

| Iteration | 0 | 30 | 60 | 90 | 114 | 151 | 180 | 210 | 241 | 271 |
|---|---|---|---|---|---|---|---|---|---|---|
| ZO-signSGD | | | | | | | | | | |
| Classified as | 8 | 8 | 8 | 8 | 0 | 0 | 0 | 0 | 0 | 0 |

| Iteration | 0 | 30 | 60 | 90 | 120 | 150 | 180 | 198 | 240 | 271 |
|---|---|---|---|---|---|---|---|---|---|---|
| ZO-M-signSGD | | | | | | | | | | |
| Classified as | 8 | 8 | 8 | 8 | 8 | 8 | 8 | 0 | 0 | 0 |

| Iteration | 0 | 40 | 80 | 120 | 160 | 200 | 240 | 280 | 320 | 349 |
|---|---|---|---|---|---|---|---|---|---|---|
| ZO-NES | | | | | | | | | | |
| Classified as | 8 | 8 | 8 | 8 | 8 | 8 | 8 | 8 | 8 | 0 |

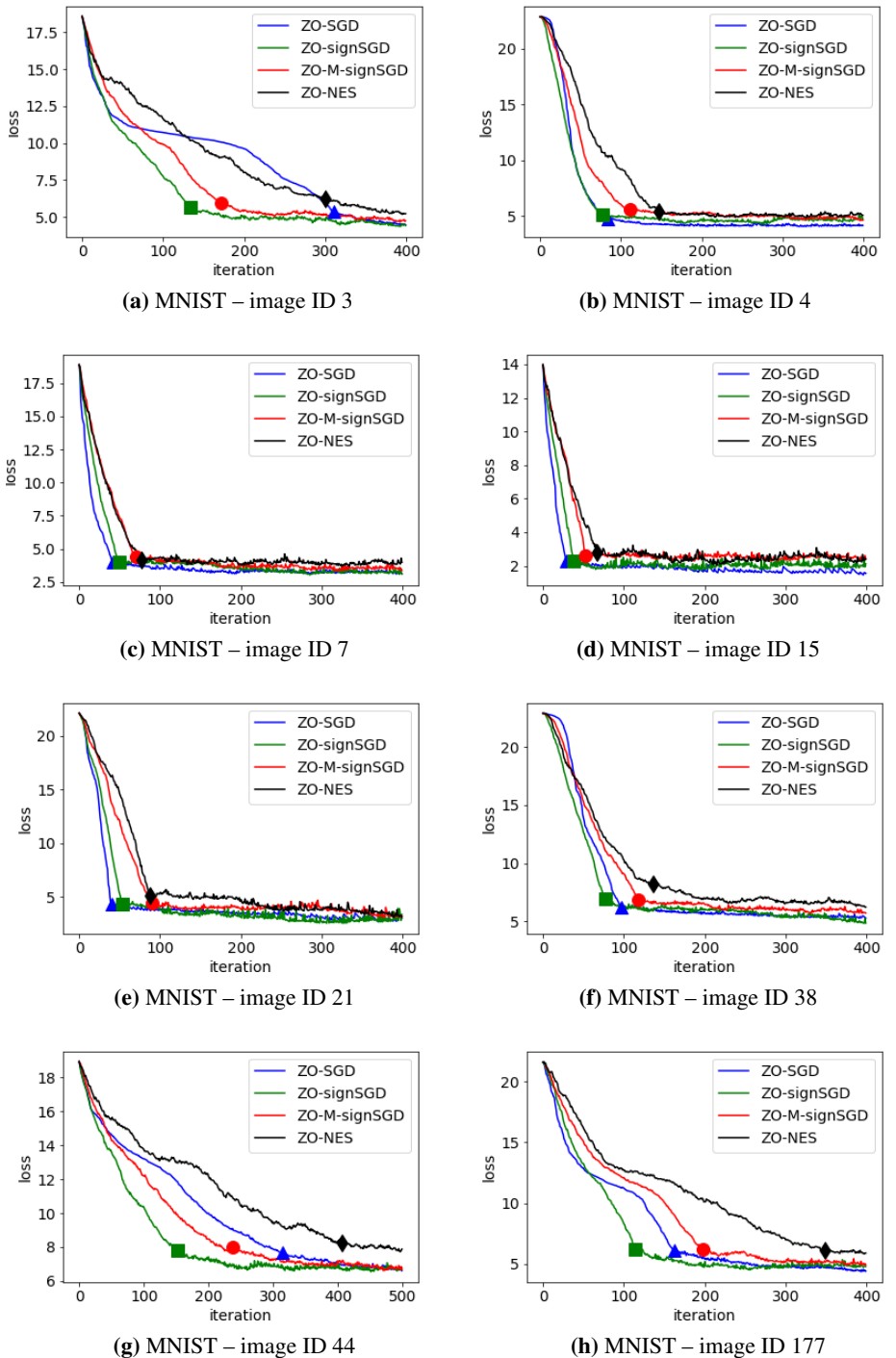

**Figure A4:** Additional plots of black-box attacking loss versus iteration on MNIST. The solid marker indicates the iteration number that finds the first successful adversarial example, and its loss corresponds to the squared $\ell_2$ distortion.

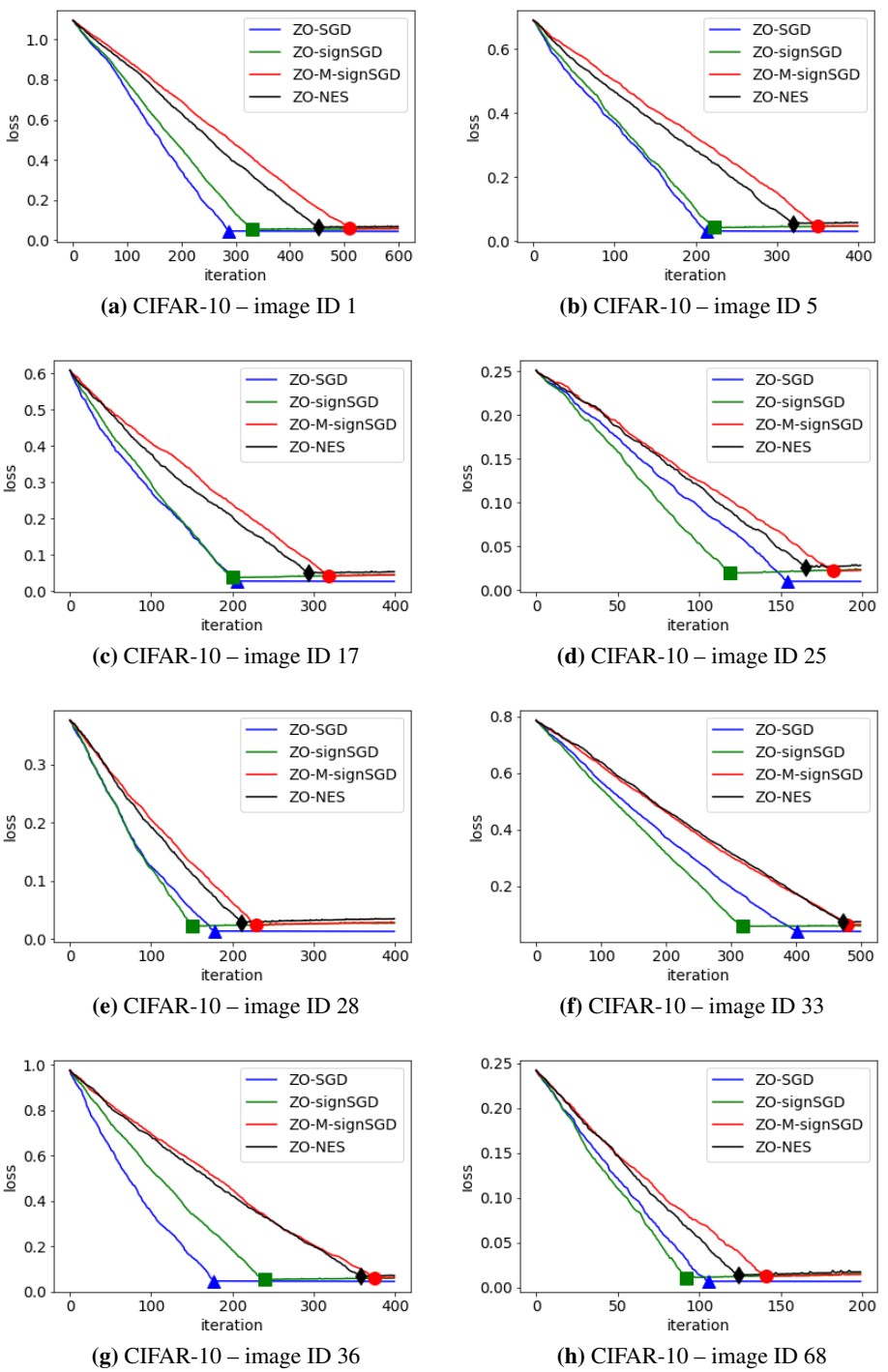

**Figure A5:** Additional plots of black-box attacking loss versus iteration on CIFAR-10. The solid marker indicates the iteration number that finds the first successful adversarial example, and its loss corresponds to the squared $\ell_2$ distortion.

