# OpenReview forum: "signSGD via Zeroth-Order Oracle"
_ICLR.cc/2019/Conference_

### Official Review · AnonReviewer2 · 2018-11-02
**Nice study of zeroth-order sign SGD**

**Rating:** 6
**Confidence:** 2

**Review:**

In this paper, the authors studied zeroth order sign SGD. Sign SGD is commonly used in adversarial example generation. Compared to sign SGD, zeroth-order sign SGD does not require the knowledge of the magnitude of the gradient, which makes it suitable to optimize black-box systems. The authors studied the convergence rate of zeroth-order sign SGD, and showed that under common assumptions, zero-order sign SGD achieves O(sqrt(d/T)) convergence rate, which is slower than sign SGD by a factor of sqrt(d). However, sign SGD requires an unrealisitcally large mini-batch size, which zeroth-order sign SGD does not. The authors demonstrated the performance of zeroth-order sign SGD in numerical experiments.

Overall, this is a well written paper. The convergence property of the zeroth-order sign SGD is sufficiently studied. The proposal seems to be useful in real world tasks.

Weaknesses:
1) out of curiosity, can we improve the convergence rate of the zeroth-order sign SGD if we assume the mini-batch size is of order O(T)? This could help us better compare zeroth-order sign SGD and sign SGD.
2) Figure 2 is too small to be legible. Also, it seems that the adversarial examples generated by zeroth-order sign SGD have higher distortion than those found by zeroth-order SGD on CIFAR-10 dataset. Is it true? If so, it would be beneficial to have a qualitative explanation of such behavior.

---

> ### Author Response · Authors · 2018-11-22
> **Response to Reviewer 2**
>
> Reviewer #2  (Q: question; R: response):
>
> We thank the reviewer for the positive comments on our paper. We provide the detailed response to each comment as below.
>
> Q: 1) out of curiosity, can we improve the convergence rate of the zeroth-order sign SGD if we assume the mini-batch size is of order O(T)? This could help us better compare zeroth-order sign SGD and sign SGD.
>
> R: Yes, the large mini-batch size of b = O(T) indeed improves the convergence rate of ZO-signSGD. As b = O(T), the convergence rate given in (9) becomes O(\sqrt{d}/\sqrt{T} + \alpha_b \sqrt{d}/\sqrt{T} + d/\sqrt{Tq}), where the last error term O(d/\sqrt{Tq}) is induced by ZO gradient estimation error. In order to further improve the rate to O(\sqrt{d}/\sqrt{T}), it is required to make the number of random direction samples $q$ proportional to $d$. Similar to other ZO methods (Liu et al. 2018; Hajinezhad et al. 2017), the large q helps to reduce the variance of ZO gradient estimates.
>
> On the other hand,  the assumption of b = O(T) might not be necessary if n < O(T), where n is the total number of individual cost functions. Suppose that b = n and we use mini-batch sampling without replacement, then ZO-signSGD becomes ZO-signGD. This leads to the convergence rate O(\sqrt{d}/\sqrt{T} + d/\sqrt{nq}). In this case, we can improve the rate to recover O(\sqrt{d}/\sqrt{T}) by only setting the number of random direction vectors induced by ZO gradient estimation, $q = O(dT/n)$. It is worth mentioning that such an improvement  cannot be achieved by ZO-signSGD using mini-batch with replacement even if b = n with the same setting of q. We refer reviewer to our detailed analysis in the last paragraph of Sec. 4.
>
> S. Liu, et al., Zeroth-order stochastic variance reduction for nonconvex optimization, NIPS, 2018
> D. Hajinezhad, et al., “Zeroth order nonconvex multi-agent optimization over networks,” arXiv preprint arXiv:1710.09997, 2017.
>
>
> Q: 2) Figure 2 is too small to be legible. Also, it seems that the adversarial examples generated by zeroth-order sign SGD have higher distortion than those found by zeroth-order SGD on CIFAR-10 dataset. Is it true? If so, it would be beneficial to have a qualitative explanation of such behavior.
>
> R: We have enlarged Figure 2. Yes, Given the first successful adversarial example, we observe that ZO-signSGD yields slightly higher L2 distortion than ZO-SGD. This is not surprising since compared to ZO-SGD, the convergence rate of ZO-signSGD involves an additional error correction term (relying on b and q in (9)). Accordingly, ZO-signSGD might converge to moderate accuracy (e.g., a solution neighborhood) rather than a very high accuracy. However, the convergence of ZO-signSGD to moderate accuracy could be much faster than ZO-SGD since the former meets a stricter convergence criterion (L2 norm of gradient) than that of ZO-SGD (squared L2 norm of gradient). We refer the reviewer to the paragraph after Eq. (9) for more discussions.
>
> In the example of generating black-box adversarial attacks, compared to convergence accuracy (in terms of attack distortion), the effectiveness of a black-box attack is measured by the number of function queries needed to achieve the first successful adversarial attack. Thus, ZO-signSGD is desired in this application due to its fast convergence to moderate accuracy. To further confirm this point, in Sec. 6 we have added an experiment to compare ZO-signSGD with a benchmark black-box attack generation method in (Ilyas et al., 2018a). Indeed, ZO-signSGD offers fast convergence to the first successful adversarial attack under limited queries.
>
> A. Ilyas, et al., Black-box adversarial attacks with limited queries and information. ICLR 2018.

---

### Official Review · AnonReviewer3 · 2018-11-03
**somewhat overclaimed and sometimes ambiguous**

**Rating:** 7
**Confidence:** 5

**Review:**

The authors proposed a zero-order version of the recent signSGD algorithm, by replacing the stochastic gradient with a usual function difference estimate. Similar convergence rates as signSGD were obtained, with an additional sqrt(d) factor which is typical in zero-order methods. Three (typical) gradient estimates based on function values were discussed. Overall, the obtained results are relatively straightforward combination of signSGD with existing zero-order techniques.

Quality: The technical part of this paper seems to be solid. The experiments, on the other hand, are quite ambiguous. First off, why do you choose that peculiar least squares binary classification problem on page 7? Is Assumption A2 satisfied for this problem? Why not use logistic regression? The experimental results are also strange: Why would ZO-signSGD converge faster than ZO-SGD or any other ZO variant? Shouldn't they enjoy similar rates of convergence? Why would taking the sign make the algorithm converge faster? Note that the original motivation for signSGD is not for faster convergence but less communication. For the second set of experiment, how do you apply ZO-SGD to generate adversarial examples? Again, why do we expect ZO-signSGD to perform better than ZO-SGD?

Clarity: This paper is mostly well-written, but the authors at times largely overclaim their contributions or exaggerate the technical challenges.
-- Page 2, 2nd line: the authors claim that "Our analysis removes the impractical assumption of b = O(T)", but in the later examples (page 6, top), they require q = O(T). How is this any different than b = O(T)? Even worse, the former case also require b = n, i.e., there is no stochasity at all...
-- Assumption A2: how crucial is this assumption for obtaining the convergence results? note that not many functions have Lipschitz continuous bounded gradients... (logistic regression is an example)
-- Page 4, top: "ZO-signSGD has no restriction on the mini-batch size b"? The rates at the end of page 5 suggests otherwise if we want the bound to go to 0 (due to the term sqrt(d/b)).
-- Page 4, top: the last two technical challenges do not make sense: once we replace f by f_mu, these difficulties go away immediately, and it is well-known how to relate f_mu with f.

Originality: The originality seems to be limited. Contrary to what the authors claimed, I found the established results to be relatively straightforward combination of signSGD and existing zero-order techniques. Can the authors elaborate on what additional difficulties they need to overcome in order to extend existing zero-order results to the signSGD case?

Significance: The proposed zero-order version of signSGD may potentially be significant in applications where gradient information is not available and yet distributed optimization is needed. This, however, is not demonstrated in the paper as the authors never considered distributed optimization.


##### added after author response #####
I appreciate the authors effort in trying to make their contributions precise and appropriate. The connection between ZO-signSGD and adversarial examples is further elaborated, which I agree is an interesting and potentially fruitful direction. I commend the authors for supplying further experiments to explain the pros and cons of the proposed algorithms. Many of the concerns in my original review were largely alleviate/addressed. As such, I have raised my original evaluation.

---

> ### Author Response · Authors · 2018-11-22
> **Response to reviewer 3 and major improvements have been made (Part I)**
>
> Response to Reviewer 3  (Q: question; R: response):
>
> Q: The obtained results are relatively straightforward combination of signSGD with existing zero-order techniques.
> And question on originality: Can the authors elaborate on what additional difficulties they need to overcome in order to extend existing zero-order results to the signSGD case?
>
> R: We are sorry to learn  that the reviewer feels our work is a relatively straightforward combination of signSGD with existing zero-order techniques. Based on the reviewer’s comments, our paper has been largely improved. In what follows, we clarity our main contributions and 'additional difficulties'.
>
> First, beyond signSGD, our established results apply to the case of mini-batch sampling without replacement. And thus, ZO-signGD can be treated as a special case in our analysis. To derive the variance of ZO gradient estimate, we require careful analysis on the effects of two types of mini-batch sampling as well as random direction sampling, and then link them with statistics of a single random gradient estimate known in the existing ZO results.
>
> Second, to derive the eventual convergence error of ZO-signSGD, we require to fill the gap between the L1 geometry of signSGD and the variance of the ZO gradient estimate in terms of squared L2 norm. Moreover, we require to study the effects of different types of ZO gradient estimators on the convergence of ZO-signSGD. In particular, sign-based gradient estimators, (11)-(12) in Sec. 5, have not been well studied in the ZO literature. These estimators can be interpreted as the ZO counterparts of first-order gradient estimators with majority vote in the centralized and distributed settings.
>
> Last but not the least, our goal is not to 'combine' ZO and signSGD. As a matter of fact, ZO-signSGD has been well motivated in the design of black-box adversarial examples (Ilyas et al., 2018a). However, the formal connection between optimization theory and adversarial ML was not fully established. Our work provides a comprehensive study on ZO-signSGD from multiple perspectives including convergence analysis, gradient estimator, and applications. We really hope that the reviewer can recognize the contributions of this work in both theory and practice.
>
>
> Q: The technical part of this paper seems to be solid. The experiments, on the other hand, are quite ambiguous. First off, why do you choose that peculiar least squares binary classification problem on page 7? Is Assumption A2 satisfied for this problem? Why not use logistic regression?
>
> R: The least squared formulation is commonly used for nonconvex machine learning (Xue et al. 2017), given the fact that the standard logistic regression yields a convex problem. Since we study ZO-signSGD in the nonconvex setting, we choose to solve the least squared binary classification problem in order to make empirical studies consistent with theory. And Assumption A2 is indeed satisfied for the proposed problem. This is not difficult to prove by the boundedness of the sigmoid function. We have clarified this point in Sec. 6.
>
> P. Xu, F. Roosta-Khorasan, and M. W. Mahoney. Second-order optimization for non-convex machine learning: An empirical study. arXiv preprint arXiv:1708.07827, 2017

---

> > ### Author Response · Authors · 2018-11-26
> > **Response to reviewer 3 and major improvements have been made (Part II)**
> >
> > Q: The experimental results are also strange: Why would ZO-signSGD converge faster than ZO-SGD or any other ZO variant? Shouldn't they enjoy similar rates of convergence? Why would taking the sign make the algorithm converge faster? Note that the original motivation for signSGD is not for faster convergence but less communication. For the second set of experiment, how do you apply ZO-SGD to generate adversarial examples? Again, why do we expect ZO-signSGD to perform better than ZO-SGD?
> >
> > R: Based on the reviewer’s comment, we realize that our explanation on the possible fast convergence of ZO-signSGD is not enough.
> >
> > First, the original motivation for signSGD is for both fast communication and fast convergence; see abstract, Sec. 3 and Figure A1 in (Bernstein et al., 2018). Thus, the motivation of signSGD is not limited to the fact that it can significantly reduce communication overhead.
> >
> > Second, it is not strange that ZO-signSGD could converge faster to at least moderate accuracy than ZO-SGD and other ZO variants. Our work, the previous work on signSGD (Bernstein et al., 2018), and many other white-box and black-box adversarial example generation methods (Goodfellow et al., 2015; Madry et al., 2018; Ilyas et al., 2018a) have shown that taking the sign could make the algorithm converge faster. We have added a subsection ‘Motivations of ZO-signSGD’ in Sec. 3 to provide rationale about why the sign operation could be beneficial to fast convergence. We repeat our discussion as below.
> >
> > “Compared to SGD-type methods, the fast empirical convergence of signSGD and ZO-signSGD has been shown in the application of generating white-box and black-box adversarial examples (Goodfellow et al., 2015; Madry et al., 2018; Ilyas et al., 2018a). As mentioned in (Bernstein et al., 2018), the sign operation could mitigate the negative effect of (coordinate-wise) gradient noise of large variance. Recall that the ZO gradient estimate is a biased approximation to the true gradient, and thus, could suffer larger noise variance than (first-order) stochastic gradients. In this context, one could benefit from ZO-signSGD due to its robustness to gradient noise. In Appendix 1, we provide two concrete examples (Fig. A1 and Fig. A2) to confirm the aforementioned analysis. In Fig. A1, we show the robustness of ZO-signSGD against sparse noise perturbation through a toy quadratic optimization problem, first introduced by (Bernstein et al., 2018). In Fig. A2, we show that gradient estimation via ZO oracle indeed encounters gradient noise of large variance. Thus, taking the sign of a gradient estimate might scale down the extremely noisy components.”
> >
> > Third, both our empirical results and theoretical results confirm that ZO-signSGD converge faster than ZO-SGD to moderate accuracy. In theory,  the convergence rate of ZO-signSGD is measured through the L2 norm | \nabla f(x_R) |_2  rather than its squared counterpart | \nabla f(x_R) |_2^2, where the latter was used to evaluate the convergence of ZO-SGD. We recall from (Ghadimi & Lan, 2013, Theorem 3.2 & Corollary 3.3) that ZO-SGD yields the convergence error E [ | \nabla f(x_R) |_2^2 ] \leq O(\sqrt{d}/\sqrt{T}).  Since  | \nabla f(x_R) |_2^2 \leq | \nabla f(x_R) |_2 as it converges, the established rate of ZO-signSGD meets a stricter convergence criterion than that of ZO-SGD. Thus, ZO-signSGD can converge faster (than ZO-SGD) to moderate accuracy, e.g., a neighborhood of a stationary point, where the size of the neighborhood is controlled by the mini-batch size b and the number of random direction vectors q. The application of black-box adversarial attack further shows that the fast convergence of ZO-signSGD to the first successful adversarial attack significantly saves the cost of function queries. We also show the superior performance of ZO-signSGD to a benchmark black-box attack generation method. We refer the reviewer to Sec. 6 for more details.
> >
> >  Ilyas, L. Engstrom, A. Athalye, and J. Lin. Black-box adversarial attacks with limited queries and information. ICLR 2018.

---

> > > ### Author Response · Authors · 2018-11-26
> > > **Response to reviewer 3 and major improvements have been made (Part III)**
> > >
> > > Q: Clarity: This paper is mostly well-written, but the authors at times largely overclaim their contributions or exaggerate the technical challenges.
> > >
> > > R: In the revised version, we have tried our best to make our claim clearer and more accurate. We answer the reviewer’s specific questions as below.
> > >
> > >
> > > Q: -- Page 2, 2nd line: the authors claim that "Our analysis removes the impractical assumption of b = O(T)", but in the later examples (page 6, top), they require q = O(T). How is this any different than b = O(T)? Even worse, the former case also require b = n, i.e., there is no stochasity at all…
> > >
> > > R: We apologize for the confusion we made in the initial version. Here the ’impractical assumption b = O(T)’ meant the assumption of i.i.d. mini-batch samples (with replacement) used in signSGD (Bernstein et al. 2018). Based on such an assumption, signSGD in (Bernstein et al. 2018) cannot cover signGD as a special case since the mini-batch of size b = n might NOT be equivalent to the entire set [n]. By contrast, our convergence analysis for ZO-signSGD applies to both mini-batch sampling schemes with and without replacement. The use of mini-batch sampling without replacement makes ZO-signSGD equivalent to ZO-signGD at b = n.
> > >
> > > Different from the mini-batch b, the number of random directions q is introduced by ZO gradient estimation. Compared to signSGD, ZO-signSGD involves an additional convergence error relying on d and q. This is the cost of gradient-free optimization methods using function difference based gradient estimates. The choice of making q proportional to d (or T) is commonly used in ZO methods (Duchi et al. 2015; Liu et al. 2018; Hajinezhad et al. 2017) to reduce the variance of ZO gradient estimates. We refer the reviewer to the last paragraph of Sec. 4 for a thorough discussion on b and q.
> > >
> > > J. C. Duchi et al., Optimal rates for zero-order convex optimization: The power of two function evaluations. IEEE TIT,  2015.
> > > S. Liu et al., Zeroth-order stochastic variance reduction for nonconvex optimization, NIPS, 2018
> > > D. Hajinezhad, et al., “Zeroth order nonconvex multi-agent optimization over networks,” 2017.
> > >
> > >
> > > Q: Assumption A2: how crucial is this assumption for obtaining the convergence results? note that not many functions have Lipschitz continuous bounded gradients... (logistic regression is an example)
> > >
> > > R: A2 is needed to bound the variance of ZO gradient estimates; see Proof of Proposition 2. The least squared form of logistic regression actually satisfies A2. We feel that A2 is not a strict assumption in nonconvex analysis, e.g., (Theorem 4, Reddi et al. 2018; Definition 3, Reddi et al. 2016). In practice, we only require that the gradient of the cost function at x_k is bounded at each time.
> > >
> > > Reddi, Sashank J., Satyen Kale, and Sanjiv Kumar. "On the convergence of adam and beyond." (2018).
> > > Reddi, S. J., Hefny, A., Sra, S., Poczos, B. and Smola, A. Stochastic variance reduction for nonconvex optimization, 2016.
> > >
> > >
> > > Q: Page 4, top: "ZO-signSGD has no restriction on the mini-batch size b"? The rates at the end of page 5 suggests otherwise if we want the bound to go to 0 (due to the term sqrt(d/b)).
> > >
> > > R: The restriction meant the assumption of i.i.d. mini-batch samples (with replacement). We wanted to emphasize that ZO-signSGD allows mini-batch sampling without replacement. The error term \sqrt(d/b) can be eliminated as b = n and when the mini-batch sampling without replacement is used. However, this is not true while using i.i.d. mini-batch samples (even if b = n). In general, the error term \sqrt(d/b) exists, which is induced by the variance of gradient estimates. We refer the reviewer to the last paragraph of Sec. 4 for a thorough discussion on the mini-batch size b.

---

> > > > ### Author Response · Authors · 2018-11-26
> > > > **Response to reviewer 3 and major improvements have been made (Part IV)**
> > > >
> > > > Q: Page 4, top: the last two technical challenges do not make sense: once we replace f by f_mu, these difficulties go away immediately, and it is well-known how to relate f_mu with f.
> > > >
> > > > R: Based on the reviewer's feedback, we have rephrased our claim and make it clearer and more accurate. About the original last two technical challenges, we would like to emphasize the following points.
> > > >
> > > > First, it is not trivial to overcome the second technical challenge, since we need to carefully investigate the effect of two mini-batch sampling schemes as well as the effect of random direction sampling on the variance of ZO gradient estimates (see Proposition 2). In particular, the use of mini-batch samples without replacement removes the assumption of i.i.d. samples. This challenge does not go away immediately even if we replace f by f_mu.
> > > >
> > > > Second, in the last technical challenge, we aimed to emphasize that both the sign operation and the ZO random gradient estimation are biased approximations to the true gradient. Note that the sign-based descent algorithm measures the convergence in L1 geometry, which introduces a mismatch with the squared L2  norm used in bounding the variance of ZO gradient estimates. In addition to relating f_mu with f, we need to translate the gradient norm from L1 to L2 and use the probabilistic convergence method to derive the eventual convergence error bound (see Theorem 1).
> > > >
> > > >
> > > > Q: Significance: The proposed zero-order version of signSGD may potentially be significant in applications where gradient information is not available and yet distributed optimization is needed. This, however, is not demonstrated in the paper as the authors never considered distributed optimization.
> > > >
> > > >
> > > > R: We agree with the reviewer that distributed optimization is an interesting setting to perform ZOsignSGD. However, even in the centralizing setting, ZO (gradient-free) methods are also attractive when the gradient is difficult or impossible to compute. For ZO-signSGD, we have shown in Sec. 3 and Sec. 6 that it is well motivated by centralization optimization problems, e.g., the design of black-box adversarial examples under limited queries.
> > > >
> > > > To further address the reviewer’s concern, we have also added a new sign-based gradient estimator (12) used for ZO distributed optimization. This results in a distributed variant of ZO-signSGD, whose convergence rate is derived in Corollary 3 and empirical performance is compared with other variants of ZO-signSGD in Figure 2. We refer the reviewer to Sec. 5 for more details.
> > > >
> > > > Hopefully, the reviewer agrees with us that our new version has been largely improved, and could re-evaluate our work towards a better score. We thanks the reviewer's efforts to review our work.

---

> > ### Comment · AnonReviewer3 · 2018-12-02
> > **more explanation of the least-squared logistic is needed**
> >
> > I find the authors' explanation for choosing the peculiar least-square logistic regression problem unsatisfactory. It appears rather contrived than what the authors claimed: "The least squared formulation is commonly used for nonconvex machine learning (Xue et al. 2017)." Note that if it is indeed commonly used, then you should be able to provide more convincing references than an arxiv paper from last year.
> >
> > The explanation "given the fact that the standard logistic regression yields a convex problem" is also problematic: why being convex is inferior? If yes, how does the least-squares logistic regression address the downside of convexity? Is there any other way?

---

> > > ### Author Response · Authors · 2018-12-02
> > > **Further explanation on least-squared logistic and will add numerical results for standard logistic regression**
> > >
> > > We thank Reviewer 3 for this insightful comment. We totally understand your concern. We did not mean that being convex is inferior, instead, we just want to show the convergence of a non-convex loss while using ZO-signSGD.  To fully address your concern, we will remove the sentence "The least squared formulation is commonly used for nonconvex machine learning (Xue et al. 2017)." And we will add experiments on solving the standard logistic regression problem once the revision window reopens.
> > >
> > > We also thank the reviewer very much for raising the evaluation score. Your previous comments really helped authors to improve the original version.

---

### Official Review · AnonReviewer1 · 2018-11-03
**Good Paper**

**Rating:** 8
**Confidence:** 3

**Review:**

The paper presents algorithms for optimization using sign-SGD when the access is restricted to a zero order oracle only, and provide detailed analysis and convergence rates. They also run optimization experiments on synthetic data. Additionally, they demonstrate superiority of the algorithm in the number of oracle calls for black box adversarial attacks for MNIST and CIFAR-10. The provided algorithm has optimal iteration complexity from a theoretical viewpoint.

The paper was, overall very well written and sufficient experiment were presented. The math also seems correct. However, I think they should have explained the motivation for the need of developing such an algorithm better. Section 3 can be improved.

I think this is an important paper because it provides a guaranteed algorithm for zero order sign-gradient descent. However, the ideas and the estimators are not novel. They show applicability of standard gradient estimators for zero order oracles for sign-sgd algorithm.

---

> ### Author Response · Authors · 2018-11-22
> **Response to reviewer 1**
>
> Response to reviewer 1(Q: question; R: response)
>
> Q: The paper was, overall very well written and sufficient experiment were presented. The math also seems correct. However, I think they should have explained the motivation for the need of developing such an algorithm better. Section 3 can be improved.
>
> R: Based on this comment, we have improved Sec. 3 and added a subsection ‘Motivations of ZO-signSGD’. Particularly, two concrete motivating examples (Appendix 1) are presented to motivate how ZO-signSGD could outperform ZO-SGD. In Fig. A1, we show the robustness of ZO-signSGD against sparse noise perturbation through a quadratic optimization problem, first introduced by (Bernstein et al., 2018). In Fig. A2, we show that ZO gradient estimates  indeed encounter gradient noise of large variance. Thus, taking the sign of a gradient estimate might scale down the extremely noisy components.
>
> Moreover, in Sec. 6, we have added an experiment to compare ZO-signSGD with a benchmark black-box attack generation method (Ilyas et al., 2018a). As we can see, ZO-signSGD offers fast convergence to the first successful adversarial attack under limited queries.
>
> Q: I think this is an important paper because it provides a guaranteed algorithm for zero order sign-gradient descent. However, the ideas and the estimators are not novel. They show applicability of standard gradient estimators for zero order oracles for sign-sgd algorithm.
>
> R: We thank R1 for the positive comments on our paper. We would like to point out that sign-based gradient estimators, e.g., (11)-(12) in Sec. 5, have not been well studied in the ZO literature. These estimators can be interpreted as the ZO counterparts of first-order gradient estimators with majority vote in the centralized and distributed settings, respectively. Here the ZO gradient estimator (12) is newly introduced for ZO distributed optimization. Even the gradient estimators (3) and (10) were used by existing ZO methods, how they affect the convergence of ZO-signSGD has not been well studied. Due to their popularity in designing black-box adversarial examples (Ilyas et al., 2018a), it is important to rigorously analyze the effect of standard gradient estimators on ZO-signSGD, in order to characterize their limitations or possible improvements.
>
> Refs: A. Ilyas, L. Engstrom, A. Athalye, and J. Lin. Black-box adversarial attacks with limited queries and information. ICLR 2018.

---

### Public Comment · (anonymous) · 2018-11-10
**Small batch convergence guarantees**

Hi Authors,

For your interest, I just want to point out another ICLR submission (from different anonymous authors):
https://openreview.net/forum?id=BJxhijAcY7.

This paper shows how small batch convergence of signSGD can be guaranteed with the additional assumption of unimodal symmetric gradient noise (e.g. Gaussian). This paper does not address the zeroth-order case.

---

> ### Author Response · Authors · 2018-11-11
> **Thanks for the reference**
>
> Thank you for pointing out the concurrent ICLR submission, which focused on the first-order Byzantine setting. The authors agreed that the extra unimodal symmetric assumption can improve the theoretical convergence bound. And indeed we showed that in the zeroth-order setting, this conclusion holds (Corollary 2). Most importantly, both papers showed the potential impact of zeroth-order and first-order signSGD on addressing practical ML problems.

---

### Author Response · Authors · 2018-11-22
**General response to all reviewers and summary of revisions**

General response to all reviewers:

We thank all reviewers for their insightful and valuable comments. Our paper has been greatly improved based on these comments. The major modifications are summarized as below.

a) In Sec. 3, we have added the subsection ‘Motivations of ZO-signSGD’ to demonstrate the possible advantages of ZO-signSGD from a high-level point of view. We then presented two concrete examples (Fig. A1 and Fig. A2) to support our intuition prior to the rigorous study on the convergence rate of ZO-signSGD.

b) In Sec. 5, we have added a new sign-based gradient estimator for ZO distributed optimization. This leads to a new variant of ZO-signSGD, whose convergence rate is illustrated in Corollary 3.

c) In Sec. 6, we have added Figure 2 to show the empirical performance of different variants of ZO-signSGD (including the new one used for ZO distributed optimization). Moreover, we compare our approach with a benchmark black-box adversarial attack method (Ilyas et al., 2018a). The new empirical results show that ZO-signSGD outperforms the benchmark in terms of both query efficiency and attack distortion.

d) Throughout the paper, we have tried our best to address reviewers’ comments and to make our presentation as clear as possible.

A. Ilyas, L. Engstrom, A. Athalye, and J. Lin. Black-box adversarial attacks with limited queries and information. ICLR 2018.

---

### Meta-Review · Area_Chair1 · 2018-12-17
**Effective approach to zero order optimization with good analysis**

**Confidence:** 5
**Recommendation:** Accept (Poster)

**Metareview:**

This is a solid paper that proposes and analyzes a sound approach to zero order optimization, covering a variants of a simple base algorithm.  After resolving some issues during the response period, the reviewers concluded with a unanimous recommendation of acceptance.  Some concerns regarding the necessity for such algorithms persisted, but the connection to adversarial examples provides an interesting motivation.